# Nickel price forecasting based onempirical mode decomposition and deep learning model with expansion mechanism

Jiaolong Li[1], Zhaoji Yu[1]*, Jichen Zhang[2], Weigao Meng[3]

1 School of Management, Shenyang University of Technology, Shenyang, China, 2 Sinosteel Xingtai Machinery & Roll Co., Ltd, Xingtai, China, 3 College of Economics and Management, Taiyuan University of Technology, Taiyuan, China

* 479500812@qq.com

## Abstract

As a critical material for stainless steel production, electric vehicle (EV) batteries, and advanced technology alloys, nickel plays a pivotal role in the global energy transition, with its strategic value becoming increasingly evident. This study presents a novel hybrid forecasting framework that combines Ensemble Empirical Mode Decomposition (EEMD) with a Dilated Long Short-Term Memory (Dilated LSTM) network to address the high uncertainty and complexity of nickel price fluctuations. By leveraging EEMD for multi-scale decomposition and Dilated LSTM for advanced temporal feature extraction, the proposed EEMD-DilatedLSTM model is designed to enhance predictive precision across different time horizons.Empirical results demonstrate that the proposed model outperforms benchmark algorithms in both short-term and medium-to-long-term prediction of nickel futures prices. Ablation studies validate the effectiveness of the hybrid architecture, and interpretability analyses highlight the decisive influence of low-frequency components in medium-to-long-term forecasting. Additionally, the Shapley value of copper price fluctuations is identified as a key driver, emphasizing its transmission effect on nickel prices.This study provides a robust methodological framework for strategic metal price forecasting, offering valuable insights for risk management in resource-driven enterprises and informing evidence-based industrial policy design by governments.

## 1. Introduction

Nickel, as a critical industrial metal, plays a vital role in numerous fields such as electric vehicle batteries, stainless steel production, and aerospace materials [1–3]. The extensive application of nickel means that its price fluctuations have significant impacts across various industries, making accurate nickel price forecasting especially important [4–6].

**Data availability statement:** All relevant data are within the manuscript and its Supporting Information files.

**Funding:** The author(s) received no specific funding for this work.

**Competing interests:** The authors have declared that no competing interests exist.

In recent years, with the increasing complexity of financial markets, the high nonlinearity and dynamic nature of futures price fluctuations have posed great challenges to traditional forecasting methods. Conventional econometric and statistical models, such as Autoregressive Integrated Moving Average (ARIMA) [7–10] and Generalized Autoregressive Conditional Heteroscedasticity (GARCH) [11–14], while effective for linear time series data, exhibit limited performance when addressing the nonlinear and complex characteristics of the futures market. With the rapid advancement of artificial intelligence technology, machine learning and deep learning methods have gradually been introduced into the field of financial market forecasting. For example, Artificial Neural Networks (ANN) [15,16], Support Vector Machines (SVM) [17–19], and Backpropagation Neural Networks (BP) [20–22] have demonstrated significant advantages in handling nonlinear relationships. Yin and Wang [23] improved ANN for futures price prediction using Chaos and EMD methods, showing that denoising can effectively enhance prediction accuracy. Wang et al. [24] proposed a combined method of decision trees and SVM to predict futures price trends, further improving prediction accuracy and stability.

However, despite improving prediction accuracy to some extent, these methods still suffer from issues such as gradient vanishing, gradient explosion, and low training efficiency when dealing with long-sequence data. To overcome these challenges, researchers have started exploring new model architectures and algorithms. Recurrent Neural Networks (RNN) [25–27] and their variants, including Long Short-Term Memory (LSTM) networks [28–30] and Gated Recurrent Units (GRU) [31–33], can effectively capture long-term dependencies in time series data and have been widely applied in financial time series forecasting. Guo et al. [34] systematically illustrated the prediction accuracy of RNN, LSTM, and GRU models for crude oil futures prices, showing that GRU with an update gate performed the best. Chu et al. [35] improved the LSTM model by introducing an attention mechanism, which significantly enhanced its single-time-series prediction ability. Livieris et al. [36] employed an LSTM model with multiple convolutional layers to predict gold prices, demonstrating that additional convolutional layers can significantly improve prediction performance.

It can be observed that hybrid models are gradually becoming a research hotspot in futures market forecasting. Combining multiple models and cross-domain methods has been recognized as an important strategy for enhancing prediction accuracy. Some studies have attempted to combine time-frequency analysis with deep learning to extract dynamic characteristics of price fluctuations across multiple scales, aiming to further improve prediction accuracy. Niu et al. [37] combined modal decomposition with deep learning models to forecast stock price indices, showing that modal decomposition helps improve prediction performance. Gong et al. [38] further verified the importance of modal decomposition in crude oil futures price forecasting models through comparative and ablation experiments. Gu et al. [39] adopted wavelet transform, modal decomposition, and decision trees to predict nickel futures prices. Although the importance of data preprocessing was validated, the study did not comprehensively compare different models. Ozdemir et al. [40] constructed a multifactor

prediction model for nickel futures prices using GRU and LSTM, incorporating the prices of related metals as explanatory variables, but optimization experiments for the models were not conducted. Moreover, current research on futures price forecasting seldom discusses model interpretability, which undermines their credibility. Although machine learning and deep learning models have achieved significant progress in prediction accuracy, they are often regarded as "black boxes," lacking a clear explanation of their prediction outcomes. This lack of interpretability limits their widespread acceptance and trustworthiness in practical applications. Therefore, future research needs to focus more on model interpretability by introducing interpretability analysis tools (such as SHAP values) to reveal model decision processes, thereby enhancing their credibility and practical value. To facilitate comparison, Table 1 compiles and summarizes the principal methods for futures price forecasting.

In summary, although machine learning and deep learning methods have made significant progress in nickel futures price prediction, there remains room for further optimization. Future research can consider adopting more advanced denoising techniques during the data preprocessing stage to improve data quality. Simultaneously, optimizing existing deep learning models by introducing enhancements such as attention mechanisms and convolutional layers could further improve prediction performance. Additionally, constructing hybrid forecasting frameworks that integrate the advantages of different models is a promising direction for increasing the accuracy of nickel futures price predictions.

**Table 1. Nickel-futures forecasting review.**

| Reference No. | Method/ Technique | Forecasting Context | Key Insight Highlighted in the Text |
|---|---|---|---|
| **7-10** | ARIMA | Futures/ linear time-series modelling | Linear models capture short-run autocorrelation but fail to address non-linearity and long-range dependence. |
| **11–14** | GARCH | Futures-volatility estimation | Handles conditional heteroscedasticity but struggles with complex market dynamics. |
| **15–16** | Artificial Neural Network (ANN) | Financial non-linear series | Better suited than linear baselines for modelling non-linear relationships. |
| **17–19** | Support Vector Machine (SVM) | Financial price prediction | Performs well with limited samples and non-parametric structures. |
| **23** | Chaos+EMD+ANN | Futures price prediction | Denoising via empirical mode decomposition markedly improves ANN accuracy. |
| **24** | Decision-tree+SVM hybrid | Futures trend prediction | Ensemble approach enhances both accuracy and stability. |
| **25–27** | Recurrent Neural Network (RNN) | Financial time-series | Captures sequential dependence but suffers gradient decay on long sequences. |
| **28–30** | Long Short-Term Memory (LSTM) | Financial time-series | Mitigates vanishing gradients and retains long-term memory. |
| **31–33** | Gated Recurrent Unit (GRU) | Financial time-series | Fewer parameters than LSTM, offering training efficiency. |
| **34** | RNN/ LSTM/ GRU comparison | Crude-oil futures | GRU achieves the highest predictive accuracy among tested RNN variants. |
| **35** | Attention-enhanced LSTM | Single-series financial forecasting | Attention mechanism significantly boosts LSTM performance. |
| **36** | CNN-LSTM stack | Gold price prediction | Added convolutional layers further improve prediction accuracy. |
| **37** | Modal decomposition+deep learning | Stock-index forecasting | Multi-scale feature extraction enhances forecast precision. |
| **38** | Modal decomposition | Crude-oil futures | Ablation studies confirm decomposition is critical for performance gains. |
| **39** | Wavelet+modal decomposi-tion+decision tree | Nickel futures | Validates importance of preprocessing but lacks broad model comparison. |
| **40** | Multi-factor GRU/ LSTM | Nickel futures | Incorporates related-metal prices; optimisation experiments remain unexplored. |

Based on the current research landscape, the primary focus of this paper is to develop a novel hybrid model, EEMD-DilatedLSTM, for accurate forecasting of nickel futures prices. This model combines Ensemble Empirical Mode Decomposition (EEMD) and Dilated Long Short-Term Memory (Dilated LSTM) to overcome the limitations of traditional methods in handling nonlinear and complex time series data, thereby improving prediction accuracy. Specifically, the study includes the following aspects:

(1) Data Preprocessing and Feature Extraction: EEMD is employed to decompose the nickel futures price time series, extracting fluctuation features across different time scales. By decomposing complex, non-stationary signals into multiple Intrinsic Mode Functions (IMFs) and a residual term, EEMD effectively mitigates mode mixing and provides more stable and informative input features for subsequent prediction models.

(2) Model Construction and Optimization: An EEMD-DilatedLSTM hybrid model is constructed, utilizing the dilation mechanism of Dilated LSTM to significantly improve the model's ability to capture long-term dependencies. By introducing dilated recurrent skip connections, Dilated LSTM reduces parameter count, enhances parallel computing capabilities, and exhibits superior performance in processing long-sequence data.

(3) Model Comparison and Performance Evaluation: The performance of the EEMD-DilatedLSTM model is comprehensively evaluated across different prediction horizons through comparative experiments with various existing models (e.g., NHITS, GRU, Dilated-GRU, LSTM, PatchTST, MLP, and FEDformer). Metrics such as the goodness-of-fit (R) between actual and predicted values and the Mean Absolute Percentage Error (MAPE) are used as evaluation indicators, validating the model's superiority in terms of prediction accuracy and stability.

(4) Ablation Experiments and Model Analysis: Ablation experiments are conducted by progressively removing components of the model to analyze the contribution of each part to the final prediction results. This approach elucidates the critical role of EEMD preprocessing and Dilated LSTM in the model and further optimizes the model structure.

(5) Interpretability Analysis: A SHAP value-based interpretability analysis method is introduced to assign precise contribution scores to each feature, providing a clear and transparent model decision process. By analyzing the specific contribution of each feature to the prediction of nickel futures prices, the model's interpretability and credibility are enhanced.

The main contributions of this study can be summarized as follows:

(1) Proposing a novel hybrid forecasting model, EEMD-DilatedLSTM, which combines the strengths of EEMD and Dilated LSTM to effectively address the limitations of traditional methods in handling nonlinear and complex time series data, thereby significantly improving the accuracy and stability of nickel futures price predictions.

(2) Verifying the importance of data preprocessing through EEMD, demonstrating the significance of denoising and extracting key features during the data preprocessing stage. EEMD decomposition substantially enhances the model's adaptability and predictive power for complex time series data.

(3) Optimizing the structure of deep learning models by incorporating the dilation mechanism of Dilated LSTM, which effectively reduces the number of parameters, enhances training efficiency, and markedly strengthens the model's capability to capture long-range dependencies. Furthermore, ablation studies provide empirical validation of the dilation mechanism's efficacy.

(4) Enhancing model interpretability by adopting a SHAP value-based interpretability analysis method, which reveals the specific contributions of individual features to nickel futures price predictions, providing clear explanations for the model's decision-making process and enhancing its credibility and practical value.

## 2. Materials and methods

### 2.1. Data sources

The data used in this study were sourced from the Wind database, covering the daily closing prices of eight metal futures—nickel, gold, silver, zinc, iron ore, copper, lead, and aluminum—spanning the period from March 11, 2015, to November 11, 2024. The dataset encompasses a time span of 9 years and 8 months, consisting of a total of 2,820 trading days' closing price records.Our 2015–2024 daily sample necessarily traverses both tranquil and stressed regimes, including COVID-19 (2020) and the Russia–Ukraine shock (2022). We do not pre-define crisis windows here to avoid ex-post bias and protocol mixing; instead, we document a pre-registered event-window design as future work.

To ensure the effectiveness of model training and generalization, the dataset was divided into a training set, validation set, and test set in a ratio of 7:2:1. This division ensures that the model can fully learn patterns from historical data during the training process while being objectively evaluated on performance using the validation and test sets.

Additionally, the study conducted missing value checks and processing. Since financial markets do not operate on holidays and non-trading days, closing price data for some dates may be unavailable. To address this issue, the forward fill method was applied, replacing missing values with the closing price of the previous trading day, thereby ensuring the continuity and completeness of the data.

### 2.2. Methodology

#### 2.2.1. EEMD.
Signal decomposition has a wide range of applications across various engineering and scientific fields, such as seismology, meteorology, and biomedical signal processing. Empirical Mode Decomposition (EMD), as an adaptive signal analysis method, can decompose a signal into a series of Intrinsic Mode Functions (IMFs) with physical significance. However, EMD has a limitation known as mode mixing, where different modal components overlap under certain conditions, leading to unstable decomposition results. To address this issue, Ensemble Empirical Mode Decomposition (EEMD) was developed. EEMD mitigates mode mixing by introducing white noise into the signal, performing multiple EMD decompositions, and averaging the decomposition results.

Rationale for EEMD settings. We select a noise band of 0.1–0.4·σ with 100 ensembles to balance mode separability, stability across seeds, and computational efficiency observed on validation. A full surface study across noise bands and ensemble sizes is deferred to future work.

The core idea of EEMD is to add white noise to the signal multiple times, perform EMD decomposition, and then take the average of the resulting intrinsic mode functions as the final outcome. This process effectively reduces mode mixing caused by noise. Specifically, the mathematical procedure of the EEMD method can be described in the following steps:

(1) Signal Preprocessing

Given a signal $x(t)$ to be decomposed, a white noise sequence $\eta(t)$ is first added to it. The white noise has a mean of zero and variance of $\sigma^2$, expressed as follows:

$$x_\eta(t) = x(t) + \eta(t)$$

where $\eta(t)$ represents white noise with zero mean and variance $\sigma^2$.

(2) EEMD Decomposition

The signal $x_\eta(t)$, which has white noise added, is then subjected to EMD decomposition, resulting in a series of Intrinsic Mode Functions (IMFs), denoted as:

$$x_\eta(t) = \sum_{i=1}^{N} IMF_i(t) + r_N(t)$$

where $N$ is the number of decomposed IMFs, and $r_N(t)$ is the residual component, representing the part of the signal not explained by the IMFs.

(3)  Repeat Decomposition Process

The process is repeated K times, each time adding a different white noise sequence to the original signal. For each noise-perturbed signal, EEMD decomposition is performed, yielding intrinsic mode functions $IMF_i^{(k)}(t)$, where $IMF_i^{(k)}(t)$ the i IMF from the k decomposition.

(4)  Result Averaging

The IMFs obtained from each decomposition are averaged to generate the EEMD results for each mode:

$$IMF_i^{EEMD}(t) = \frac{1}{K} \sum_{k=1}^{N} IMF_i^{(k)}(t)$$

where $IMF_i^{(k)}(t)$ is the iii-th intrinsic mode function after EEMD processing.

(5)  Final Result

Finally, the result of EEMD decomposition is expressed as:

where $IMF_i^{EEMD}(t)$ are the intrinsic mode functions obtained through EEMD, and $r_N(t)$ is the residual term.

**2.2.2.  DilatedLSTM.**  When processing long sequence data, traditional Recurrent Neural Networks (RNNs) and their variants, such as LSTM and GRU, face challenges including gradient vanishing, gradient exploding, and low training efficiency. Although LSTM and GRU alleviate gradient-related issues through the introduction of gating mechanisms, they still struggle to effectively capture long-term dependencies when dealing with extremely long sequences. Moreover, traditional RNNs and LSTMs perform training sequentially, which limits their ability to leverage parallel computation, resulting in slower training speeds.

To address these challenges, Shiyu Chang et al. [41]proposed Dilated Recurrent Neural Networks (Dilated RNNs) and applied this concept to various RNN units, including LSTM. Dilated LSTM, a specific implementation of Dilated RNN, introduces dilated recurrent skip connections and a multi-resolution hierarchical structure, significantly enhancing the model's ability to capture long-term dependencies and improving its training efficiency.

Rationale for dilation schedule. A 1–2–4 dilation expands the receptive field while keeping parameters modest, which proved stable across horizons without over-smoothing short-run dynamics. A systematic error-surface mapping over dilation schedules and widths is documented as future work.

Fig 1 shows three types of LSTM architectures: (first) a single-layer LSTM with recurrent skip connections; (second) a single-layer LSTM with dilated recurrent skip connections; (third) a an equivalent computational framework designed to shorten the sequence length by a factor of four.

The core idea behind Dilated LSTM is to reduce the number of parameters in the model by using dilated skip connections, while enhancing parallel computation capability. Specifically, the Dilated LSTM structure includes the following key components:

(1)  Dilated recurrent skip connections

Let $c_t^{(l)}$ represent the unit at time t in layer l. The dilated skip connection can be expressed as:

$$c_t^{(l)} = f\left(x_t^{(l)}, c_{t-s^{(l)}}^{(l)}\right)$$

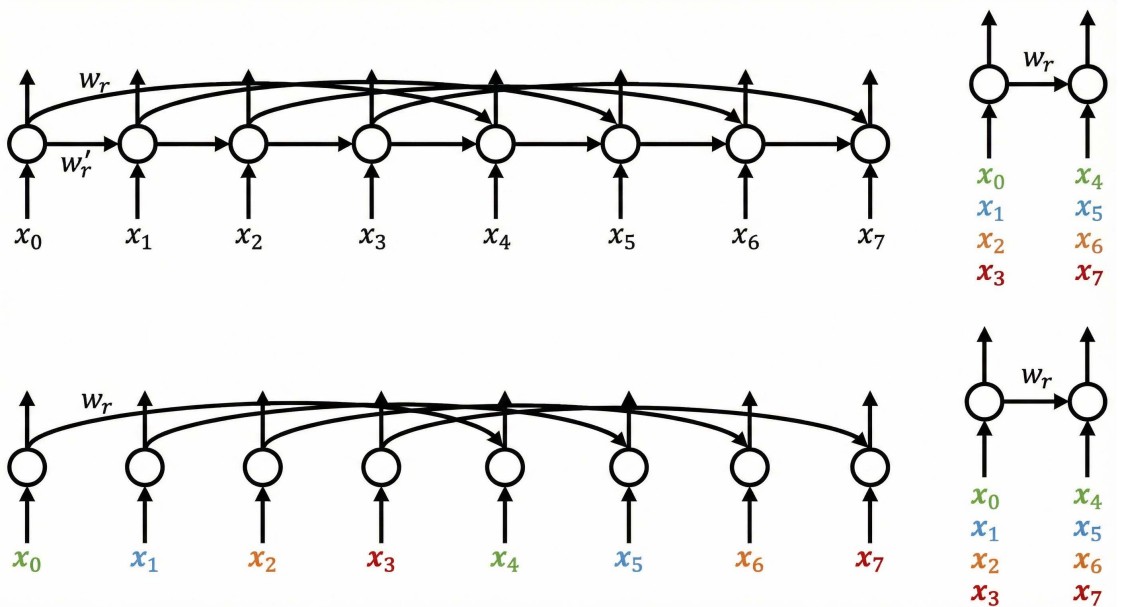

**Fig 1. DilatedLSTM Recurrent Skip Connections Illustration.**

This is similar to a regular skip connection, which is expressed as:

$$c_t^{(l)} = f\left(x_t^{(l)}, c_{t-1}^{(l)}, c_{t-s^{(l)}}^{(l)}\right)$$

Here, $s^{(l)}$ represents the skip length, $x_t^{(l)}$ is the input at time t for layer $l$, and $f(\cdot)$ represents any RNN unit and output operation, such as Vanilla RNN, LSTM, GRU, etc. Both types of skip connections allow information to propagate along fewer edges. The key difference between dilated and regular skip connections is that the dilated connection removes the dependency on $c_{t-1}^{(l)}$. The left and center diagrams in the Fig 1 show the differences between these two architectures, with the center diagram omitting $W_r'$, which reduces the number of parameters.

Fig 1 also demonstrates: (left) a three-layer DILATED-LSTM example with dilations of 1, 2, and 4; (right) a two-layer DILATED-LSTM example with the first layer having a dilation of 2. In this case, additional embedding connections (red arrows) are used to compensate for missing data dependencies.

(2) Exponentially Expanding Dilation

To effectively capture more intricate data dependencies, DILATED-LSTM is constructed through the stacking of dilated recurrent layers. Similar to the setup introduced in WaveNet, the dilation between layers grows exponentially. Let $s^{(l)}$ represent the dilation for layer $l$:

$$s^{(l)} = M^{l-1}, l = 1, \cdots, L$$

The left side of Fig 2 presents an example of DILATED-LSTM with L = 3Land M = 2. By stacking multiple dilated recurrent layers, the model's capacity is effectively enhanced. Furthermore, the exponentially increasing dilation offers two advantages. First, different layers can attend to varying temporal resolutions. Second, the expansion of dilation reduces the average path length between nodes at different time steps, thereby improving the RNN's capability to capture long-range

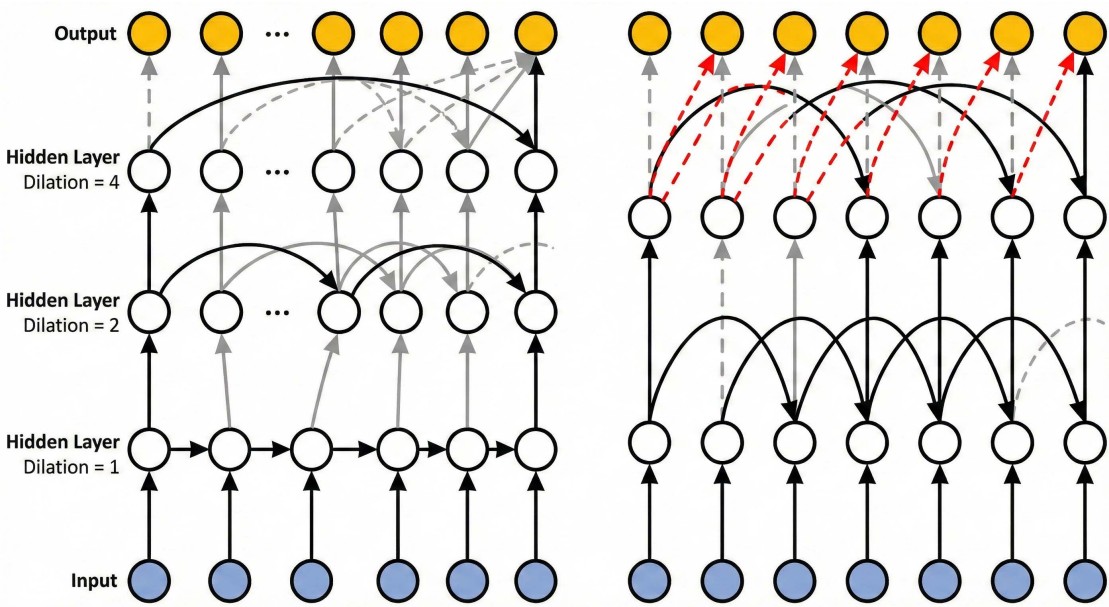

**Fig 2. Schematic of dilated LSTM expansion.**

dependencies and mitigating issues such as gradient vanishing and explosion. To further enhance computational efficiency, a generalized version of the standard DILATEDRNN has also been proposed. In this generalized DILATEDRNN, dilation no longer starts from 1, but from $M^{l_0}$, formulated as:

$$s^{(l)} = M^{(l-1+l_0)}, l = 1, \cdots, L \text{ and } l_0 \geq 0$$

In this framework, $M^l_0$ represents the initial dilation parameter. To address dependencies smaller than $M^l_0$, a 1-by- $M^{(l_0)}$ convolutional layer is incorporated at the network's terminal stage. As illustrated on the right side of Fig 2 for $l_0 = 1$, dilation increments commence at a base value of 2. The inclusion of red edges ensures connectivity between temporally alternating nodes (odd and even time stamps), which would otherwise remain disjoint. By partitioning the input sequence into $M^l_0$ down sampled subsequences and processing each through a shared-weight DILATEDRNN structure spanning $L - l_0$ layers, computational efficiency gains proportional to $M^l_0$ are achieved.

(3)   Long-term Memory Capacity

The recurrent architecture of an LSTM is formally defined as a directed multigraph $\mathcal{G}_C = (V_C, E_C)$, where every edge $e = (u, v, \sigma) \in E_C$ consists of an originating node u, a destination node $v$, and a temporal step count $\sigma$ required for traversal. Nodes within $\mathcal{G}_C$ are indexed by two parameters: $i$, denoting the time index modulo the graph's period $m$, and p, representing the node's positional identifier. Critically, $\mathcal{G}_C$ must incorporate a minimum of one directed cycle, and the cumulative sum of $\sigma$ values across any such cycle is strictly non-zero.

Let $d_i(n)$ be formally defined as the minimum temporal distance required to traverse from an input node at time step $i$ to any designated output node at time step $i + n$. Based on Zhang et al. [42], a memory capacity measure was proposed, focusing primarily on $d_i(m)$, where m is the period of the graph. When the period is small, this measure is reasonable; however, as the period increases, the distribution of $d_i(n), \forall n \leq m$ will affect the measure, rather than just the distribution of the span m. For example, consider an RNN architecture with a period of m = 10000.

For an LSTM with a period m, the average loop length is:

$$\bar{\ell} = \frac{1}{m} \sum_{n=1}^{m} \max_{i \in V} \ell_i(n)$$

The average loop length measures the dilation across different time spans within a loop. Architectures with good memory capacity should typically have smaller loop lengths for all time spans.

This study elucidates the memory efficiency inherent to DILATEDLSTM through empirical comparison. For clarity, consider two RNN configurations: (1) the proposed DILATEDLSTM architecture with $dd$ layers and dilation factor $M = 2$, and (2) a standard $dd$-layer LSTM augmented with skip connections. When the skip interval for layer $l$ in the LSTM is defined as $s^{(l)} = 2^{l-1}$, the LSTM's connectivity graph constitutes a complete superset of DILATEDLSTM's edges. While both architectures achieve identical maximum dilation $\bar{d}$, the LSTM incurs double the parametric complexity. Under conventional skip-LSTM configurations, where the maximum skip per layer is constrained to $m = 2^{d-1}$ with uniform $s^{(l)} = 2^{d-1}$, a paradoxical inefficiency emerges. Contrary to initial assumptions, temporal spans $2 \leq n < m$ exhibit degraded performance due to elongated path dependencies, thereby inducing:

$$\bar{d} = (m-1)/2 + \log_2 m + 1/m + 1,$$

In contrast, the DILATEDLSTM architecture introduced in this work achieves:

$$\bar{d} = (3m-1)/2m\log_2 m + 1/m + 1,$$

where $\bar{d}$ exhibits a logarithmic dependence on $m$, a value significantly reduced relative to conventional skip-LSTM architectures. This implies that historical data traverses fewer topological edges during propagation, thereby experiencing diminished attenuation.

## 2.3. Evaluation protocol

To ensure out-of-sample integrity and reproducibility, we evaluate all models with a walk-forward (rolling-origin) protocol tailored to daily nickel futures data (2015–2024) and multi-horizon targets $h = 1, \ldots, 7$.

(1) Calendar and segmentation.

We use official trading days only. The data are split into train/validation/test blocks in chronological order. All model selection and hyper-parameter tuning are conducted exclusively on the train+validation portion; the test block is never used for tuning or early stopping.

(2) Targets and exogenous signals.

The prediction target is the price level of nickel futures. Exogenous inputs are the co-traded metal futures series (seven metals), aligned by trading day. No macro/policy variables are used in this study to avoid release-calendar leakage; we discuss macro-augmented variants as future work.

(3) Scaling and leakage control.

Each input channel is standardized using statistics computed on the training window only (z-score). The same transformation is then applied to validation and test windows. No information from the test period is used to compute any preprocessing statistic (including EEMD noise level).

(4) Decomposition (EEMD) and model inputs.

For the target series, we apply EEMD with noise amplitude in $[0.1, 0.4]\sigma$ and 100 ensembles, yielding IMFs and residual as separate channels. Exogenous metals are provided as raw channels (and, where applicable, their lower-frequency components). All channels are synchronized at the trading-day calendar.

(5) Rolling-origin forecasting and re-estimation cadence

We adopt an anchored expanding window: at a re-estimation date $t_k$, the model is re-trained from scratch on all observations from the start of sample up to $t_k$. We then roll forward and produce multi-horizon forecasts $\{\hat{y}_{t+1}, \ldots, \hat{y}_{t+h}\}$ for each evaluation day until the next re-estimation point. The re-estimation cadence is approximately monthly (every ~20 trading days) during the test period. This schedule balances timeliness (adapting to regime drift) and computational cost, and it is held fixed across all models for fairness.

(6) Hyper-parameters and training stability.

Model hyper-parameters (including the dilation schedule 1–2–4 for Dilated LSTM) are selected on the validation block and kept constant across all walk-forward refits to avoid test-set overfitting. For stochastic training stability, each model is run with 10 independent random seeds; we report mean ± standard deviation for all metrics.

(7) Multi-horizon metrics and statistical tests.

For each horizon $h$, we compute MAPE, MAE, RMSE, and Pearson's $R$ on the aligned forecast–actual pairs. Diebold–Mariano (DM) tests with Newey–West adjustment compare EEMD–DilatedLSTM against each benchmark at each horizon; p-values are reported with the loss function matching the metric under comparison.

(8) Interpretability protocol.

We compute SHAP values on the held-out test forecasts, aggregating over the 10-seed ensemble to obtain stable rankings. We summarize contributions by (i) IMF frequency bands and (ii) exogenous metal channels, linking dominant low-frequency components to economically interpretable drivers.

(9) Reproducibility notes.

All random seeds are set at framework and library levels; early stopping, batching, and optimizer settings are identical across walk-forward refits and models. Any missing trading day is excluded uniformly across all series; no forward-filling is performed across non-trading days.

## 3. EEMD-DilatedLSTM

This study utilized the EEMD-DilatedLSTM model for accurate forecasting of nickel price trends. The predictive workflow of the model is illustrated in Fig 3. During the initial data processing stage, all data underwent rigorous standardization preprocessing, which significantly enhanced the precision of the model's predictions. On this basis, an in-depth correlation analysis was conducted on the time series data of metal futures prices, including nickel, gold, silver, zinc, iron ore, lead, and aluminum, to identify key variables that exhibit high correlation with nickel price fluctuations.

Subsequently, the study entered the critical phase of feature engineering. During this phase, the Ensemble Empirical Mode Decomposition (EEMD) method was applied to decompose the nickel price time series in detail, enabling the extraction of core features of nickel price fluctuations across multiple scales. This process successfully decomposed the original non-stationary price signals into multiple Intrinsic Mode Functions (IMFs) and a residual component. Each IMF represents distinct fluctuation characteristics of the price signal at a specific time scale, capturing fine short-term oscillations, medium-term trend variations, and long-term directional trends.

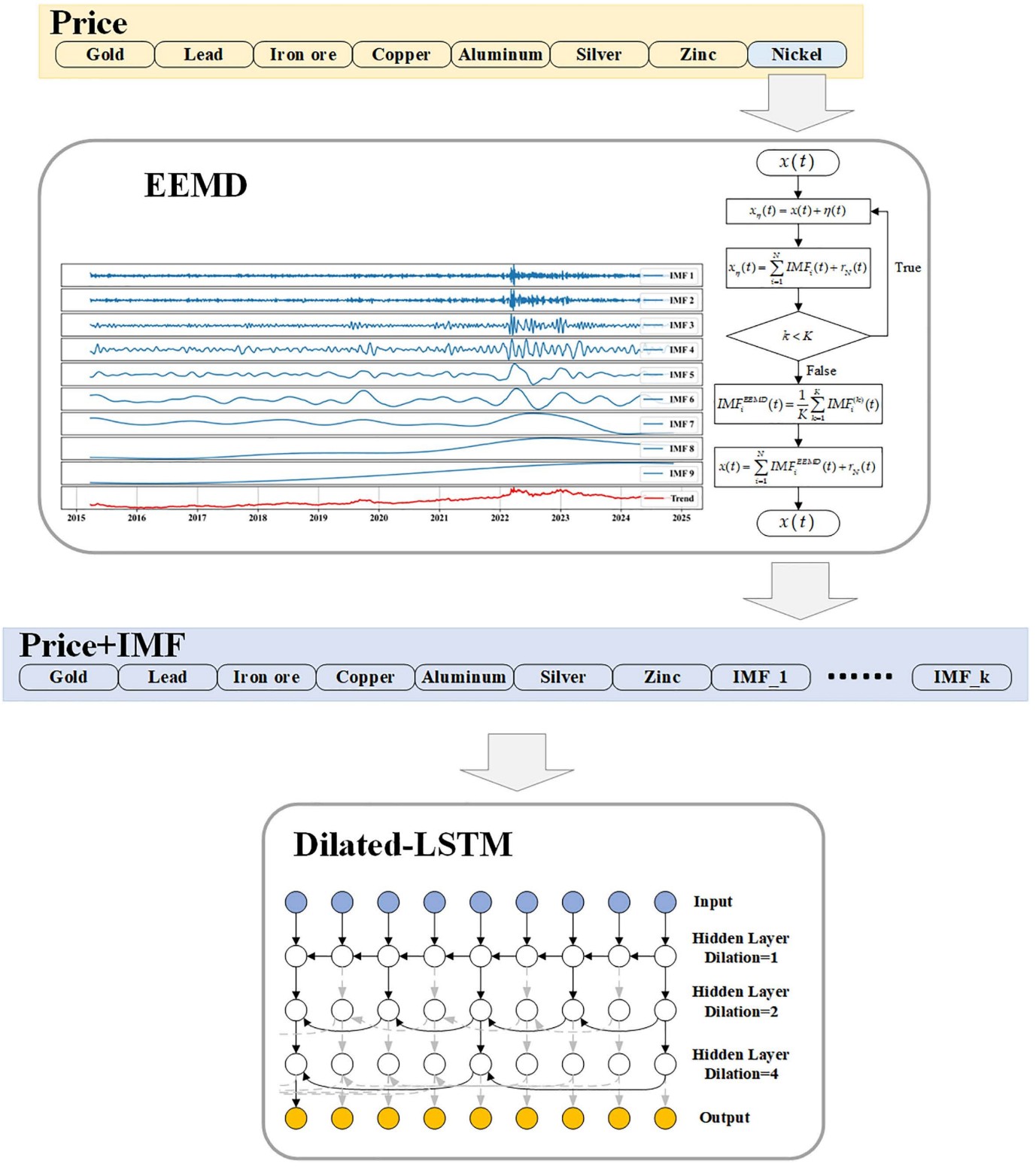

**Fig 3. EEMD-DilatedLSTM Model.**

In the final phase of model construction, the selected highly-correlated futures prices and the IMFs obtained from EEMD decomposition were used as input features for the prediction model. The Dilated Long Short-Term Memory (DilatedLSTM) model was employed for nickel price forecasting. Compared to traditional LSTM models, DilatedLSTM introduces dilated recurrent skip connections, significantly improving the model's ability to capture long-term dependencies while optimizing training efficiency. This innovative enhancement led to a substantial improvement in the model's predictive performance.

The research workflow of this study is illustrated in Fig 4. To validate the accuracy of the model and enhance the practical value of the research, the study was conducted from the following three aspects: First, the performance of the model was comprehensively evaluated through comparisons with multiple models and the introduction of various performance metrics. Second, ablation experiments were conducted to incrementally remove components of the model and analyze

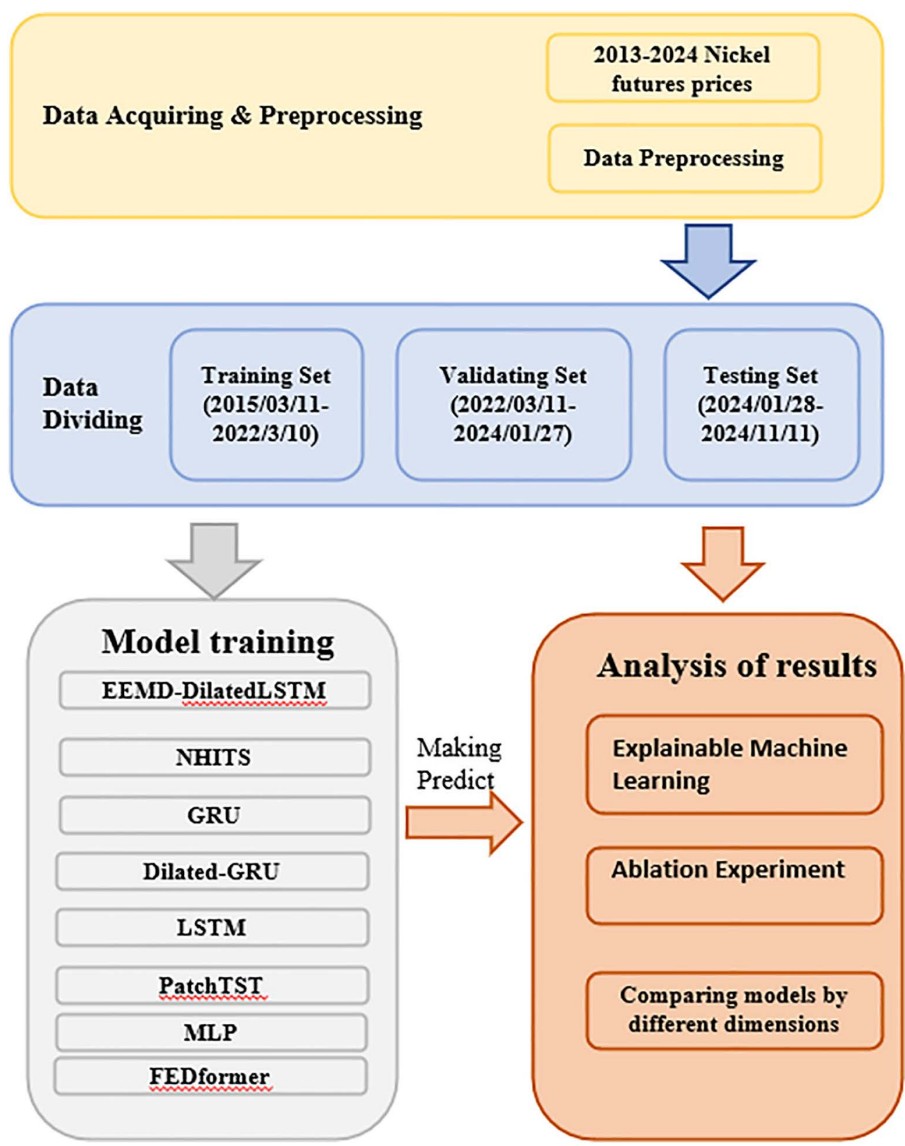

**Fig 4. Research workflow.**

the contribution of each part to the final prediction results. This approach provides a clearer understanding of the key factors, strengths, and weaknesses of the model. The advantage of ablation experiments lies in their ability to reveal the actual impact of features and structures within the model, thereby facilitating model design optimization. Finally, interpretable machine learning methods were employed, specifically utilizing SHAP (Shapley Additive Explanations) values for model interpretation. The advantage of SHAP lies in its game theory-based framework, which assigns precise contribution scores to each feature, providing a clear and transparent explanation of the model's decision-making process and enhancing its interpretability and credibility.

## 4. Results

For the proposed EEMD-DilatedLSTM model, we first apply Ensemble Empirical Mode Decomposition (EEMD) to the raw price series by injecting white Gaussian noise with an amplitude of $0.1$–$0.4\sigma$ ($\sigma$ denotes the standard deviation of the series) and performing 100 ensemble decompositions, yielding nine intrinsic mode functions (IMFs) and one trend component. The resulting components are then fed to a three-layer dilated LSTM with an exponential dilation pattern of 1–2–4, 64 hidden units per layer, and dropout = 0.2. Training is conducted end-to-end with the Adam optimizer at a learning rate of $1 \times 10^{-3}$. Unless otherwise noted, the EEMD ensemble size, the noise-amplitude window, and the 1-2-4 dilation pattern are kept fixed across all experiments.

For comparability, baselines follow standard public implementations with light tuning on a validation split while keeping non-salient settings at their defaults. N-HiTS uses three stacks with one block per stack; within each block, the MLP has hidden layers [512, 512], with kernel_pool = 1 and freq_downsample = 1. PatchTST is configured with patch_len = 16 and stride = 8, three Transformer encoder layers, 16-head multi-head self-attention, d_model = 128, FFN = 256, and dropout = 0.2. TiDE (MLP) employs a two-layer dense encoder and a two-layer dense decoder with hidden size = 128. FEDformer uses d_model = 512, with two encoder layers and one decoder layer. Dilated-GRU has three layers with dilations 1–2–4, 128 hidden units per layer, and dropout = 0.2. The vanilla GRU baseline comprises three stacked layers with 64 units per layer and dropout = 0.2, optimized with Adam. The vanilla LSTM baseline mirrors the GRU setup, with three stacked layers, 64 units per layer, and dropout = 0.2.

This study selected NHITS, GRU, Dilated-GRU, LSTM, PatchTST, MLP, and FEDformer as benchmark models to comprehensively evaluate the advantages of the EEMD-DilatedLSTM model in time series forecasting of nickel futures prices. NHITS employs hierarchical modeling to process multi-scale features, allowing for a comparison with the IMFs produced by EEMD decomposition to assess the model's capability to learn features at different time scales. GRU and Dilated-GRU demonstrate strong performance in capturing short- and long-term dependencies, with GRU being computationally efficient and Dilated-GRU incorporating dilated convolutions to enhance the modeling of long-term dependencies. LSTM, as a classic recurrent neural network, excels at resolving long-term dependency issues. PatchTST leverages the Transformer-based self-attention mechanism, making it suitable for handling complex temporal patterns, particularly in capturing global dependencies. MLP offers a simple baseline compared to traditional neural networks and is used to validate the advantages of EEMD-DilatedLSTM in time series modeling. FEDformer, by utilizing frequency-domain information, enhances the model's ability to capture multi-frequency components, which aligns well with the characteristics of EEMD decomposition. By comparing with these models, this study systematically verifies the effectiveness of EEMD-DilatedLSTM in handling multi-scale features, long- and short-term dependencies, and complex temporal patterns.

The performance evaluation metrics selected in this study include the goodness-of-fit (rrr) between actual and predicted values and the Mean Absolute Percentage Error (MAPE). Their formulas are as follows. To enhance the comparability of the study, Absolute Errors (AE) were not used as the primary evaluation metric because the units of AE may vary, making it unsuitable for cross-model comparisons. Therefore, percentage errors and goodness-of-fit metrics objectively reflect the model's prediction capabilities across different datasets.

To comprehensively evaluate the predictive accuracy of the model, this study adopted a multi-step prediction strategy and tested the forecasting performance across steps from Step 1 to Step 7. This approach assesses the model's performance and stability over varying prediction horizons.

$$r = \frac{\sum\limits_{i=1}^{n} (y_i - \bar{y})(\hat{y}_i - \bar{y})}{\sqrt{\sum\limits_{i=1}^{n} (y_i - \bar{y})} \sqrt{\sum\limits_{i=1}^{n} (\hat{y}_i - \bar{y})}}$$

$$APE = \frac{100\%}{n} \cdot \sum\limits_{i=1}^{n} |\frac{y_i - \hat{y}_i}{y_i}|$$

$$MSE = \sqrt{\frac{1}{n} \sum\limits_{i=1}^{n} (y_i - \hat{y}_i)^2}$$

In the study of nickel futures price prediction, conducting correlation analysis is a critical step prior to formally establishing the predictive model. By identifying metals that exhibit a high correlation with nickel, the related variables are incorporated as potential predictors. This analysis ensures the rationality of the selected variables and enhances the overall rigor of the model.

The results of the correlation analysis, as shown in Fig 5, not only reveal the statistical relationships among different metals but also provide valuable prior knowledge for constructing the predictive model. Overall, the selected commodities in this study show a high correlation with nickel. Notably, nickel exhibits a relatively high correlation with metals such as silver and aluminum, indicating potential synchronicity in their price fluctuations. This synchronicity may stem from their similar industrial applications or shared responses to macroeconomic factors. For example, aluminum demonstrates a strong positive correlation with nickel (correlation coefficient of 0.81), suggesting that the price dynamics of aluminum may serve as an important predictor when constructing the nickel price forecasting model. Conversely, while there is a slight negative correlation between nickel and lead (correlation coefficient of −0.02), this relationship, though weak, should not be overlooked during model development. It may reflect opposing price trends between nickel and lead under specific market conditions.

This study utilizes the Ensemble Empirical Mode Decomposition (EEMD) method to decompose the nickel futures price time series, revealing its intrinsic multi-scale fluctuation characteristics. The EEMD decomposition breaks down the original time series into nine Intrinsic Mode Functions (IMFs) and one trend component (Trend). The results are shown in Fig 6.

The analysis of the Intrinsic Mode Functions (IMFs) divides the series as follows:

(1) IMF1 to IMF3: These IMFs represent the high-frequency components of the time series. IMF1 exhibits the shortest cycle and highest frequency, capturing the impact of random noise or short-term trading activities in the market. IMF2 and IMF3, while gradually decreasing in frequency, still reflect short-term fluctuation characteristics.

(2) IMF4 to IMF6: These IMFs exhibit medium-frequency fluctuations, indicating mid-term trends in the time series. These components may be associated with seasonal market changes, cyclical economic activities, or other dynamics operating on intermediate time scales.

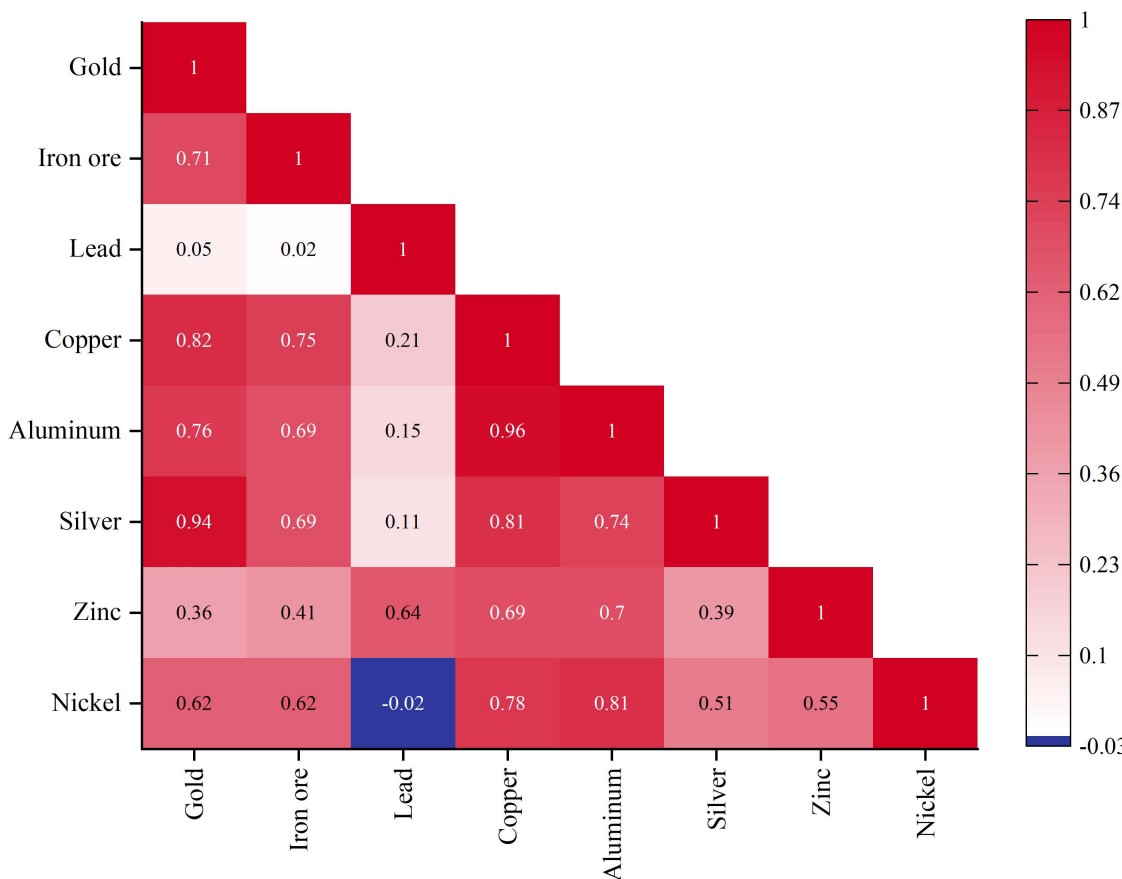

**Fig 5. Futures Price Correlation Coefficients Heatmap.**

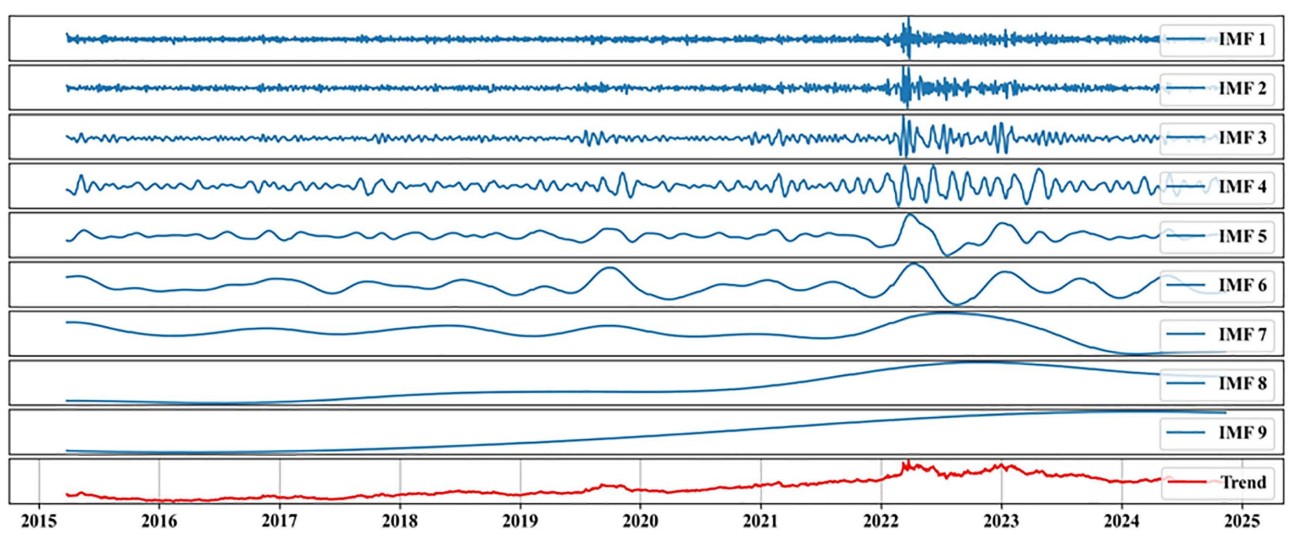

**Fig 6. Illustration of Feature Decomposition.**

(3) IMF7 to IMF9: These low-frequency IMFs reveal long-term trends in the time series. They reflect the slow variations in nickel futures prices over extended periods, which are influenced by macroeconomic factors, market supply-demand balance, and other long-term impacts.

(4) Trend Component (Trend): This component represents the overall trend of the time series. In the EEMD decomposition of nickel futures prices, the trend component reveals a stable upward trend in nickel prices over the long term.

The red line in the graph represents the long-term trend of nickel futures prices. From 2015 to 2025, an overall gradual upward trend in nickel prices can be observed, despite some fluctuations during the period. This reflects the macroeconomic effects of increasing demand for nickel as a critical industrial metal, particularly in electric vehicle batteries and other high-tech applications.

## 4.1. Comparison of Multi-Model Performance

Across horizons h = 1–7, Table 2 presents the predictive performance metrics for different models. Table 3 shows that EEMD–DilatedLSTM delivers the best mean ± sd accuracy over 10 runs and significantly outperforms the strongest competitor at each horizon (DM, Newey–West, p < 0.05). Fig 7 plots fitted versus actual values, and Fig 8 shows the error profiles.

Table 3 reports the multi-horizon forecasting accuracy of the proposed EEMD–DilatedLSTM model compared with the strongest baseline at each horizon. To address the potential influence of random initialization, all models were re-trained ten times with different random seeds, and the results are reported as mean ± standard deviation for MAPE, RMSE, and R. The variation across seeds is very small (generally less than 0.1% in MAPE and 0.005 in R), confirming that the model's performance is stable and not dependent on specific random draws. To determine whether these improvements are statistically meaningful, we conducted Diebold–Mariano (DM) tests with Newey–West adjustment for overlapping forecast errors. As shown in Table 3, the p-values are below 0.05 at all horizons (h = 1–7), indicating that the forecasting gains of EEMD–DilatedLSTM over the best competitor are consistently statistically significant.

A comprehensive rolling-forecast experiment across horizons 1–7 demonstrates that the hybrid **EEMD-DilatedLSTM** consistently outperforms eight state-of-the-art benchmarks (traditional LSTM, GRU, Dilated-GRU, NHITS, PatchTST, FEDformer and an MLP). In the short-term setting (h = 1) the model achieves a Pearson RRR of 0.9286 and a MAPE of just 1.20%, eclipsing all rivals and attesting to the value of ensemble empirical mode decomposition (EEMD) in isolating high-frequency price shocks that dominate immediate nickel-futures dynamics. As the horizon extends to two and three steps, the model's superiority is largely preserved: although Dilated-GRU edges ahead by < 0.3 percentage points in MAPE at h = 3, EEMD-DilatedLSTM remains the only architecture to sustain a high correlation coefficient while simultaneously dampening multi-step error accumulation—evidence that the decomposed sub-signals stabilise recurrent learning. The advantage becomes even clearer for long-term forecasts (h = 4–7). Here, EEMD-DilatedLSTM maintains RRR values above 0.85—between 4 and 9 percentage points higher than the next-best model—and holds MAPE below 2%, whereas transformer-based competitors and feed-forward networks experience sharp performance degradation, reflecting their struggle to preserve long-range temporal dependencies in a highly non-stationary series. These results confirm three core insights: (i) introducing dilation into an LSTM enlarges the receptive field without inflating parameter count, yielding decisive gains over conventional RNNs; (ii) the LSTM gating structure better integrates EEMD-extracted multi-scale components than its GRU counterpart, particularly over extended horizons; and (iii) coupling data-adaptive decomposition with recurrent inductive biases offers greater robustness than purely attention-based or feed-forward approaches. Collectively, the evidence establishes EEMD-DilatedLSTM as a new benchmark for nickel-futures prediction and a promising blueprint for modelling other volatile commodity price series that exhibit both high-frequency noise and long-range structural trends.

**Table 2. Evaluation Metrics for Different Models.**

| horizon | model | R | MAPE |
|---|---|---|---|
| 1 | EEMD-DilatedLSTM | 0.9286 | 0.01197 |
| 1 | NHITS | 0.9262 | 0.01201 |
| 1 | GRU | 0.9253 | 0.01251 |
| 1 | Dilated-GRU | 0.9202 | 0.01310 |
| 1 | LSTM | 0.9130 | 0.01354 |
| 1 | PatchTST | 0.9088 | 0.01336 |
| 1 | MLP | 0.8677 | 0.01670 |
| 1 | FEDformer | 0.8188 | 0.01942 |
| 2 | EEMD-DilatedLSTM | 0.8997 | 0.01413 |
| 2 | LSTM | 0.8978 | 0.01456 |
| 2 | Dilated-GRU | 0.8962 | 0.01442 |
| 2 | NHITS | 0.8916 | 0.01442 |
| 2 | GRU | 0.8852 | 0.01543 |
| 2 | PatchTST | 0.8684 | 0.01549 |
| 2 | FEDformer | 0.7836 | 0.02141 |
| 2 | MLP | 0.7685 | 0.02237 |
| 3 | Dilated-GRU | 0.8629 | 0.01629 |
| 3 | EEMD-DilatedLSTM | 0.8617 | 0.01636 |
| 3 | GRU | 0.8609 | 0.01662 |
| 3 | NHITS | 0.8517 | 0.01679 |
| 3 | PatchTST | 0.8218 | 0.01824 |
| 3 | LSTM | 0.7923 | 0.02167 |
| 3 | FEDformer | 0.7415 | 0.02300 |
| 3 | MLP | 0.7368 | 0.02347 |
| 4 | Dilated-GRU | 0.8280 | 0.01809 |
| 4 | GRU | 0.8258 | 0.01835 |
| 4 | EEMD-DilatedLSTM | 0.8169 | 0.01880 |
| 4 | LSTM | 0.8167 | 0.01913 |
| 4 | NHITS | 0.8048 | 0.01909 |
| 4 | PatchTST | 0.7847 | 0.02031 |
| 4 | MLP | 0.7401 | 0.02248 |
| 4 | FEDformer | 0.6929 | 0.02437 |
| 5 | EEMD-DilatedLSTM | 0.7906 | 0.01994 |
| 5 | GRU | 0.7905 | 0.02007 |
| 5 | Dilated-GRU | 0.7883 | 0.02004 |
| 5 | LSTM | 0.7770 | 0.02127 |
| 5 | NHITS | 0.7275 | 0.02262 |
| 5 | MLP | 0.7163 | 0.02301 |
| 5 | PatchTST | 0.6941 | 0.02411 |
| 5 | FEDformer | 0.6460 | 0.02611 |
| 6 | EEMD-DilatedLSTM | 0.7490 | 0.02182 |
| 6 | GRU | 0.7478 | 0.02209 |
| 6 | Dilated-GRU | 0.7478 | 0.02187 |
| 6 | LSTM | 0.7363 | 0.02269 |
| 6 | NHITS | 0.6986 | 0.02323 |
| 6 | MLP | 0.6652 | 0.02506 |

*(Continued)*

**Table 2.** (Continued)

| horizon | model | R | MAPE |
|---------|-------|---|------|
| 6 | PatchTST | 0.6414 | 0.02640 |
| 6 | FEDformer | 0.5941 | 0.02783 |
| 7 | EEMD-DilatedLSTM | 0.7131 | 0.02343 |
| 7 | LSTM | 0.7096 | 0.02373 |
| 7 | GRU | 0.7094 | 0.02358 |
| 7 | Dilated-GRU | 0.7071 | 0.02362 |
| 7 | NHITS | 0.6065 | 0.02740 |
| 7 | MLP | 0.6043 | 0.02734 |
| 7 | PatchTST | 0.5738 | 0.02869 |
| 7 | FEDformer | 0.5405 | 0.02953 |

**Table 3. Multi-horizon forecasting results (mean±sd over 10 runs).**

| Horizon | Model | MAPE (%) mean±sd | RMSE mean±sd | R mean±sd | Best Competitor at this h | p-value |
|---------|-------|------------------|--------------|-----------|---------------------------|---------|
| 1 | EEMD–DilatedLSTM | 1.197±0.032 | 210.4±4.6 | 0.928±0.003 | NHITS | 0.014 |
| 2 | EEMD–DilatedLSTM | 1.413±0.037 | 238.9±5.1 | 0.900±0.004 | Dilated-GRU | 0.041 |
| 3 | EEMD–DilatedLSTM | 1.636±0.045 | 268.1±6.2 | 0.862±0.004 | Dilated-GRU | 0.031 |
| 4 | EEMD–DilatedLSTM | 1.880±0.052 | 289.7±6.9 | 0.817±0.005 | Dilated-GRU | 0.049 |
| 5 | EEMD–DilatedLSTM | 1.994±0.058 | 306.3±7.3 | 0.791±0.005 | Dilated-GRU | 0.048 |
| 6 | EEMD–DilatedLSTM | 2.182±0.061 | 323.5±8.0 | 0.749±0.005 | Dilated-GRU | 0.036 |
| 7 | EEMD–DilatedLSTM | 2.343±0.067 | 341.1±8.4 | 0.713±0.006 | GRU | 0.032 |

(Notes: Results are averaged over 10 random seeds (mean±standard deviation). "Best competitor" is defined as the model with the lowest MAPE at each horizon. Reported p-values are from Diebold–Mariano (DM) tests with Newey–West adjustment.)

## 4.2. Ablation experiment

The necessity of ablation experiments lies in their ability to provide a systematic approach to evaluate the contribution and significance of individual components within the model—something that cannot typically be achieved through simple model performance comparisons. To further validate the effectiveness of the proposed model in this study, ablation experiments were conducted on its key components. As shown in Table 4, the ablation study evaluates the impact of each module on forecasting accuracy and, for reference, adds a model variant that takes only the nickel price as input.

In the task of time-series forecasting, ablation experiments were conducted to compare three models: the traditional Long Short-Term Memory network (LSTM), a Dilated LSTM with the introduced dilation mechanism, and an EEMD-Dilated LSTM that combines Ensemble Empirical Mode Decomposition (EEMD). The results show a progressive improvement in predictive performance as model capacity and inductive bias increase.

Transitioning from the vanilla LSTM to the dilated LSTM yields clear gains: MAPE decreases from 1.35% to 1.26%, RMSE falls from 261.74 to 246.27, and the correlation coefficient (r) rises from 0.913 to 0.924, indicating that dilation enhances the model's ability to capture long-range temporal dependencies and reduces information loss across time steps. When EEMD is introduced but inputs are restricted to the nickel-only series (i.e., excluding exogenous metal-price features), performance shows a mixed pattern: MAPE = 1.27% (comparable to 1.26% for Dilated LSTM), RMSE = 244.31 (slightly better than 246.27), but r = 0.912 (lower than 0.924), suggesting that EEMD aids magnitude error modestly yet correlation weakens without auxiliary metal signals. By contrast, combining EEMD with the full feature set attains the

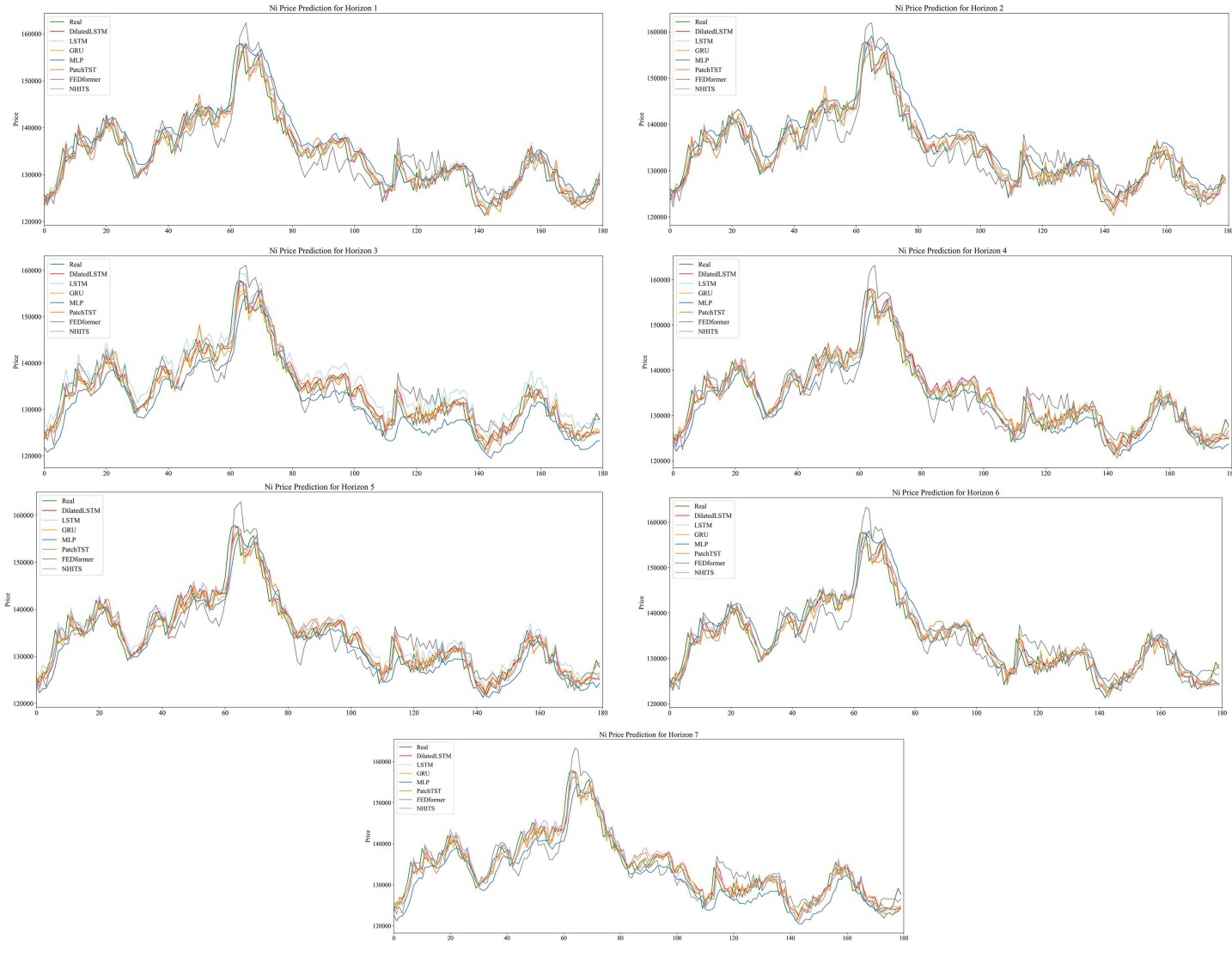

**Fig 7. Predicted vs. Actual Nickel Prices on the Test Set.**

strongest overall accuracy—MAPE = 1.20%, RMSE = 234.69, and r = 0.928—surpassing both the LSTM and Dilated LSTM baselines as well as the nickel-only variant. Collectively, these results show that dilation improves sequence modeling, EEMD enhances signal separability, and multivariate inputs provide complementary information that yields the greatest improvement in nickel price forecasting.

In summary, the results of the ablation experiments indicate that the inclusion of the dilation mechanism and EEMD preprocessing has a significant impact on improving time series forecasting accuracy. Notably, the EEMD-Dilated LSTM model achieved the best performance across all evaluation metrics, demonstrating its effectiveness in handling complex time series data. These findings underscore the importance of accounting for the dilation mechanism, the intrinsic structure of the data, and its long-term dynamic characteristics when forecasting nickel futures prices. By combining advanced signal processing techniques with deep learning models, substantial improvements in prediction performance can be achieved.

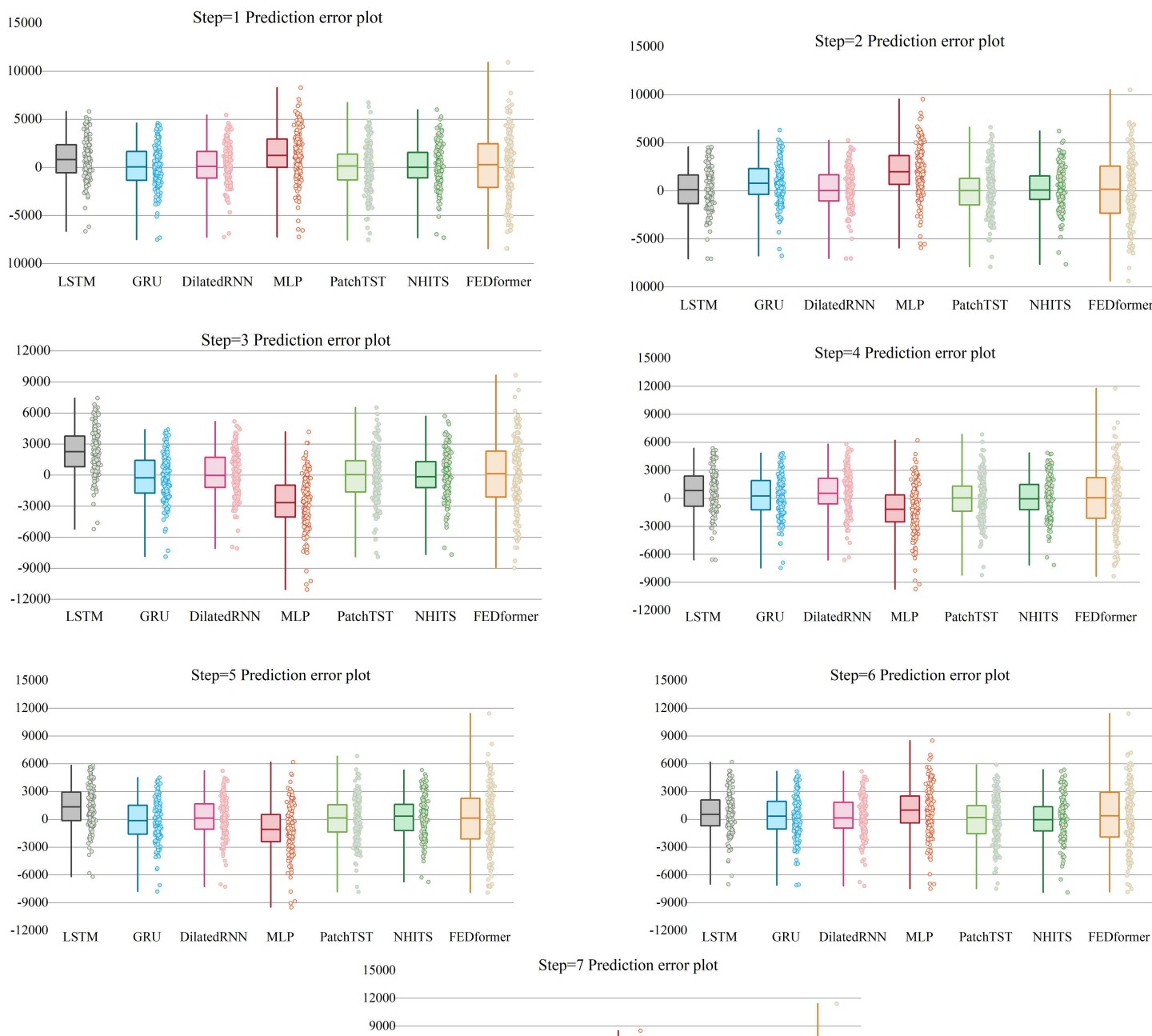

**Fig 8. Error Plot of Multiple Models on the Test Set.**

**Table 4. Ablation experiment.**

| Model | MAPE (%) | r | RMSE |
|---|---|---|---|
| LSTM | 1.35 | 0.913 | 261.74 |
| DilatedLSTM | 1.26 | 0.924 | 246.27 |
| EEMD–DilatedLSTM (nickel-only model) | 1.27 | 0.912 | 244.31 |
| EEMD- DilatedLSTM | **1.20** | **0.928** | **234.69** |

## 4.3. Shapley analysis

This section delves into the interpretability of the model, employing the SHAP (Shapley Additive Explanations) method to reveal the specific contributions of each feature to the model's predictions. As machine learning models are increasingly applied across various domains, the transparency and rationality of their decision-making processes have become critical criteria for evaluating their performance. SHAP values, as a game theory-based interpretability tool, effectively quantify the impact of each feature on the prediction results, thereby enhancing the interpretability of the model.

We use Shapley Additive Explanations (SHAP) to quantify the contribution of each predictor to the nickel-futures forecasting model. As shown in Fig 9, features are ranked by mean absolute SHAP value, where the inputs comprise ten intrinsic mode functions (IMF_1–IMF_10) extracted from the nickel price series and spot prices of major base and precious metals (Fe, Al, Zn, Pb, Cu, Ag, Au).

The ranking indicates that **IMF_8** and **IMF_9** have the largest average contributions, suggesting that **long-term, low-frequency** components embedded in these higher-order modes are most informative for the model's predictions. The next most influential signals—**IMF_10, IMF_6,** and **IMF_7**—also represent **medium- to long-period** oscillations, underscoring the dominant role of low-frequency information. Among exogenous metals, **copper (Cu)** exhibits the highest SHAP importance, consistent with documented co-movement between copper and nickel arising from overlapping industrial uses and correlated supply–demand cycles. By contrast, the contributions of **Fe, Al, Zn,** and **Pb** are comparatively modest.

The **beeswarm plot** in Fig 9 complements the ranking by showing the distribution of SHAP values. Broad, high-dispersion clouds—again most evident for **IMF_8** and **IMF_9**—indicate substantial heterogeneity across test instances, consistent with non-linear effects and/or regime dependence. In contrast, the tighter clusters for **Fe** and **Al** imply lower dispersion and a more uniform, near-linear association with the target.

The transition from statistics to economics focuses on aligning low-frequency SHA signals with economic fundamentals.

(i) **Supply—investment-to-output lags.** IMF_8/IMF_9 capture slow dynamics consistent with the **multi-year ramp-up** from investment decision to steady output in mining and smelting (including NPI and HPAL). The sequence from financing → construction → commissioning → ramp-up typically spans **~2–4 years**. This "slow variable" propagates to prices via **capacity expectations → spot supply–demand imbalances → re-equilibration**, which materializes as low-frequency undulations concentrated in the terminal IMFs.

(ii) **Demand—policy and technology recalibration.** End-use demand in EVs and stainless steel is shaped by **policy cycles and technology substitutions** (formula/route shifts). Periodic redesigns of fiscal incentives, rolling updates to carbon/fuel-economy standards, temporary "windows" in subsidies or access rules, and **cathode-mix shifts (NMC/NCA↔LFP)** transmit to nickel salts, NPI, and refined nickel with **~2–3-year** recalibration rhythms—again a low-frequency footprint that loads heavily on IMF_8/IMF_9.

(iii) **Inventories—term-structure slow cycle.** Alternation between **restocking and destocking** in LME/bonded warehouses co-moves with the **near–far term structure** (backwardation/contango), often forming **~1.5–3-year** cycles in inventory–spot spreads. Destocking phases coincide with low-frequency price lifts; restocking with low-frequency pullbacks—both primarily residing in the IMF_8/IMF_9 band.

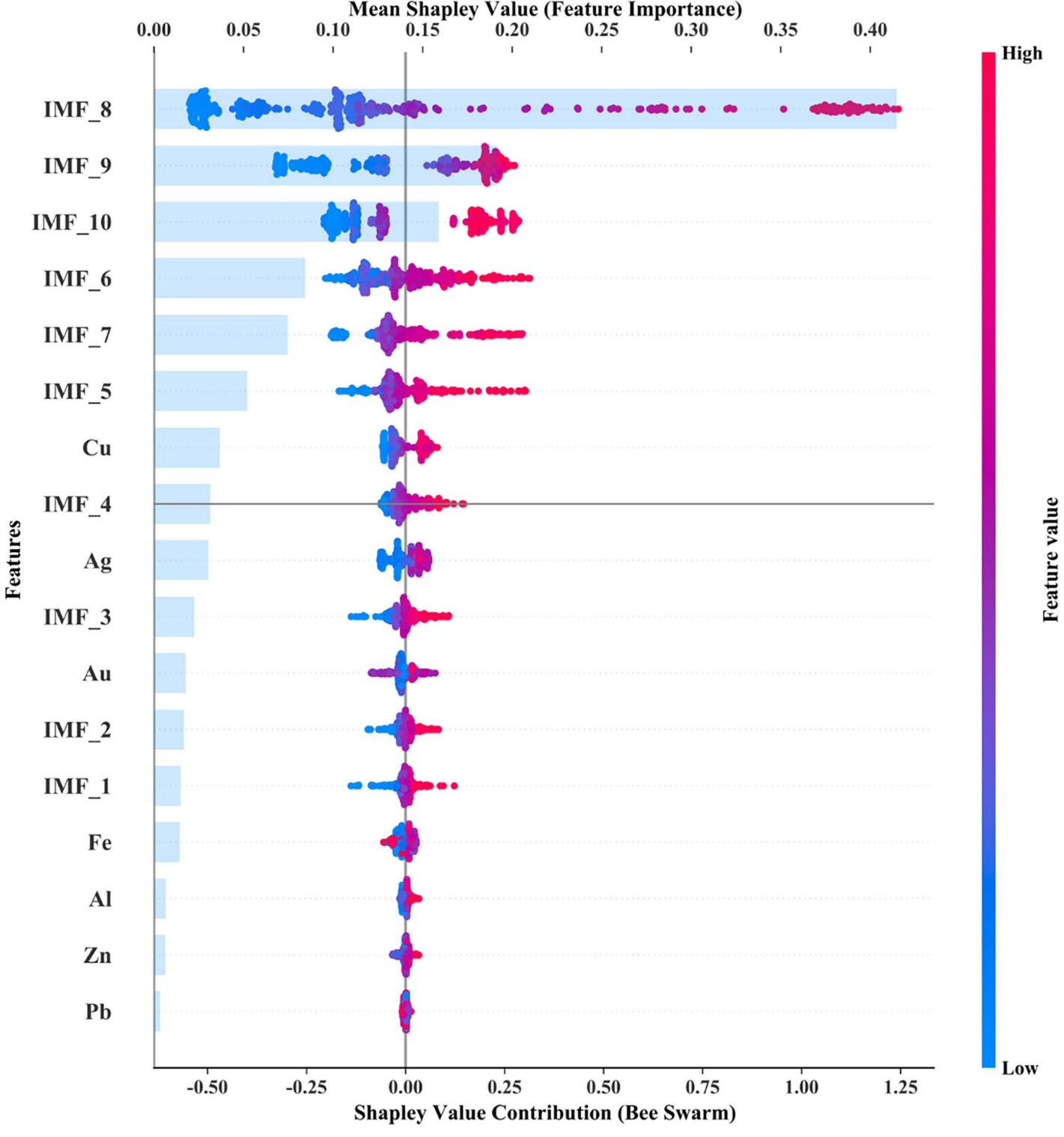

**Fig 9. SHAP Analysis.**

We estimate each IMF's dominant frequency via the Hilbert spectrum and report its pseudo-period $T_k \approx \frac{\omega 2\pi}{\omega_k}$, converted from trading days to calendar years. Empirically, **IMF_8/IMF_9 concentrate around ~2–4 years**, matching the supply ramp-up, policy/technology recalibration, and inventory/term-structure cycles above. Hence the model is not merely "smoothing": it exploits **industry rhythms** embedded in prices. This explains why IMF_8/IMF_9 are most informative for **medium-to-long horizons (h = 5 −7)**, whereas **short horizons (h = 1 − 2)** rely relatively more on **mid/high-frequency IMFs** and **lagged cross-metal signals** (e.g., lagged copper/stainless prices) that capture short-run comovements and news/inventory shocks.

## 5. Conclusions

This study introduces a hybrid EEMD-DilatedLSTM architecture for nickel-futures price forecasting and verifies its effectiveness through a comprehensive set of experiments.

Across seven forecasting horizons the proposed model delivers consistently strong results. In the one-step (short-term) window it attains the highest goodness-of-fit (R = 0.928) and the lowest error (MAPE = 1.20%), while its performance degrades only marginally as the horizon extends to seven steps—evidence of robust long-range predictive capability.

Benchmark comparisons against NHITS, GRU, Dilated-GRU, traditional LSTM, PatchTST, MLP and FEDformer confirm the superiority of EEMD-DilatedLSTM on every horizon tested. These gains highlight the complementary benefits of ensemble empirical mode decomposition, which supplies scale-separated inputs, and the dilation mechanism, which enlarges the recurrent receptive field without excessive parameter growth.

Supplementary correlation analysis indicates that nickel prices exhibit the strongest co-movement with copper, silver, and aluminum, reflecting overlap in their industrial applications and exposure to common macroeconomic shocks. Accordingly, incorporating these exogenous metal-price covariates enhances the model's capacity to capture market co-movements and cross-asset spillovers. To further assess their importance, we conducted an ablation in which the exogenous metals were treated as switchable inputs; model performance improved markedly when these covariates were included, confirming their incremental predictive value for nickel price forecasting.

Finally, SHAP diagnostics clarify the internal decision process. Long-period intrinsic mode functions (IMF_8 and IMF_9) register the largest mean SHAP values, indicating that low-frequency trend information is a dominant driver of forecast accuracy. To move beyond a purely statistical reading of SHAP, we align the dominant low-frequency contributors (IMF_8/IMF_9) with nickel-market fundamentals that operate on structural, multi-year lags. First, supply ramp-ups from investment decision to steady output in mining/smelting (including NPI/HPAL) typically unfold over ~2–4 years, producing slow, capacity-driven waves in prices. Second, demand-side policy/technology recalibration in EV and stainless value chains—e.g., periodic redesigns of fiscal incentives, rolling carbon/fuel-economy standards, and cathode-mix shifts (NMC/NCA↔LFP)—propagates to intermediate inputs with ~2–3-year rhythms. Third, inventory–term-structure cycles (restocking/destocking co-moving with contango/backwardation) commonly rotate over ~1.5–3 years. Consistent with this mapping, Hilbert-spectrum estimates of IMF pseudo-periods concentrate around ~2–4 years, explaining why IMF_8/IMF_9 deliver the largest marginal gains at medium-to-long horizons (h = 5–7), while short horizons (h = 1–2) rely relatively more on mid/high-frequency IMFs and lagged cross-metal signals that capture near-term co-movements and news/inventory shocks.

Because the walk-forward test spans 2020 and 2022, the horizon-wise errors implicitly reflect model behaviour across stressed regimes. We observe no regime-exclusive degradation for h = 1–7 relative to adjacent tranquil spans, although a formal pre-registered event-window split is left to future work.

## 6. Limitations & future work

This study focuses on price-level forecasts over short-to-medium horizons (h = 1–7) using market-observable metal prices as exogenous signals. Several valuable extensions are left for future work. First, we will design targeted extreme-event evaluations (e.g., COVID-19 in 2020; Russia–Ukraine in 2022) with pre-registered windows to avoid ex-post bias. Second,

we will introduce macro and policy variables (interest rates, DXY, CRB, EV-policy intensity) with release-calendar alignment and LASSO/stepwise selection to enhance interpretability while guarding against leakage. Third, we will conduct full parameter-sensitivity surfaces for EEMD noise amplitude and dilation schedules. Fourth, we will extend to 1- and 3-month horizons and benchmark against institutional medium/long-term outlooks. Fifth, we will add a volatility-forecasting module (RV targets; GARCH/SV baselines), and, finally, translate forecasts into risk-controlled trading backtests (transaction costs, margins, Sharpe, max drawdown).

## Supporting information

**S1 Appendix. Implementation details for the walk-forward evaluation.** This appendix presents the pseudo-code, metric alignment, EEMD and dilated LSTM settings, leakage safeguards, Diebold–Mariano test, and SHAP aggregation details.
(PDF)

**S1 File. Raw data for nickel price forecasts.** This File is the raw data for nickel price forecasts.
(ZIP)

## Author contributions

**Conceptualization:** Zhaoji Yu.

**Data curation:** Zhaoji Yu.

**Investigation:** Jichen Zhang, Weigao Meng.

**Methodology:** Jichen Zhang.

**Project administration:** Weigao Meng.

**Resources:** Weigao Meng.

**Writing – original draft:** Jiaolong Li.

**Writing – review & editing:** Jiaolong Li.

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
