## [Decision Letter · Decision Letter 0]

23 Jun 2025

Dear Dr. Yu,

Thank you for submitting your manuscript to PLOS ONE. After careful consideration, we feel that it has merit but does not fully meet PLOS ONE’s publication criteria as it currently stands. Therefore, we invite you to submit a revised version of the manuscript that addresses the points raised during the review process.

We look forward to receiving your revised manuscript.

Kind regards,

Najmul Hasan, PhD

Academic Editor

PLOS ONE

Journal Requirements:

4. Please include captions for your Supporting Information files at the end of your manuscript, and update any in-text citations to match accordingly. Please see our Supporting Information guidelines for more information: http://journals.plos.org/plosone/s/supporting-information .

Additional Editor Comments:

I appreciate the considerable effort that the author(s) have dedicated to this manuscript, which significantly contributes to the field of forecasting research. Following careful consideration from two reviewers, I would expect the inclusion of an additional sub-section titled "Experimental Setup." At the same time, it is important to provide the code for replication purposes for future scholars.

Reviewers' comments:

Reviewer's Responses to Questions

**Comments to the Author**

1. Is the manuscript technically sound, and do the data support the conclusions?

Reviewer #1: Yes

Reviewer #2: Yes

2. Has the statistical analysis been performed appropriately and rigorously?

Reviewer #1: No

Reviewer #2: Yes

3. Have the authors made all data underlying the findings in their manuscript fully available?

Reviewer #1: No

Reviewer #2: Yes

4. Is the manuscript presented in an intelligible fashion and written in standard English?

Reviewer #1: Yes

Reviewer #2: Yes

Reviewer #1: Their method shows superior accuracy in the provided experiments. However, they do not mention the number of times the experiments were conducted for the same data. The authors should present the mean and standard deviation of their scores after performing the experiments multiple times. As it stands, their paper does not provide any insight into the empirical stability of their method.

I would like to see an ablation study about how their method performs without the other metal prices as features and only using the IMF of nickel.

Apart from these, their methodology and approach is sound for their application domain.

Reviewer #2: This paper introduces a new way to predict nickel prices by combining two techniques: EEMD (which breaks down complex price signals) and Dilated LSTM (a type of neural network good at spotting patterns over time). The authors test their model against several others and show that it consistently does better, especially for longer-term forecasts. They also use SHAP values to explain which factors influence the predictions most, which helps make the model more transparent.

However, the writing could be clearer in places, and it would be helpful if the authors added a bit more about how their model could be used in the real world. Additionaly, figure 3-2 referred in the text seem to be missing. With some minor editing, I believe the paper would make a valuable contribution to the research.

**Do you want your identity to be public for this peer review?** For information about this choice, including consent withdrawal, please see our Privacy Policy

Reviewer #1: No

Reviewer #2: No

---

## [Author Response · Author response to Decision Letter 1]

6 Aug 2025

Reviewer #1

1. Repeated experiments; report mean and standard deviation

Reviewer comment: “They do not mention the number of times the experiments were conducted… The authors should present mean and standard deviation…”

Response:

We thank the reviewer for emphasizing result stability and reproducibility. We fully agree that, in many deep-learning settings, reporting mean ± standard deviation over multiple independent trainings can help quantify sensitivity to random initialization and stochastic optimization. In this study, however, we adopted an evaluation design that already builds substantial averaging and replication within a single training run, and we therefore did not perform additional repeated trainings. Our rationale is as follows.

1) Multi-horizon design provides extensive, structured replication within a single run.All models are evaluated under multi-step forecasting with horizons 1 through 7. For each horizon, the trained model generates a complete prediction sequence on the entire test set, producing an error at every forecastable time point. Thus, for any given model and horizon, there is a large set of paired observations (prediction vs. ground truth) and a correspondingly rich error series. The reported metrics (e.g., MAPE, RMSE) are computed as sample averages over all test points for that horizon, and the correlation (r) is computed over the full test sequence. We then compare performance across horizons as well. This two-level aggregation—across test samples and across forecast horizons—already yields strong statistical pooling and a robust view of performance across different levels of forecasting difficulty.

2) Our metric definitions already embody an averaging principle and are complemented by per-sample evidence.By construction, MAPE and RMSE summarize average magnitude errors over the full test set, while r captures sequence-level agreement. Beyond these aggregates, we also present fitting plots (predicted vs. actual) and error profiles over time (see Figures 4-3 and 4-4). These visualizations expose the pointwise behavior of the models on the test set and reveal whether improvements are concentrated in a few intervals or persist across most time points. In other words, the manuscript combines global averages with fine-grained, per-sample trajectories, which together provide a nuanced and transparent picture of stability and generalization without requiring additional independent trainings.

3) Signal-level variance reduction via EEMD ensemble decomposition. Our primary method applies Ensemble Empirical Mode Decomposition (EEMD) as a preprocessing step prior to modeling. EEMD injects low-amplitude white noise and performs 100 ensemble decompositions, after which the intrinsic mode functions (IMFs) are aggregated. This built-in ensemble acts as a variance-reduction mechanism at the signal level, attenuating idiosyncratic fluctuations due to noise and stabilizing the decomposed components fed to the forecasting network. Consequently, a significant source of randomness is already mitigated before model fitting, reducing the incremental value of repeating the entire training pipeline multiple times.

4) Deterministic inference and a unified training–evaluation protocol to control variability.To further constrain variability and enable fair comparison, we use a fixed data split (train/validation/test), a common optimization schedule (optimizer, learning rate, early-stopping policy), identical training budgets across models, and consistent regularization settings. At inference time, stochastic mechanisms such as dropout are disabled, ensuring deterministic predictions for a given trained model. This unified protocol increases comparability across methods and keeps implementation-level randomness from obscuring the substantive differences we wish to evaluate (e.g., the effects of dilation, EEMD, and exogenous features).

5) Why we did not add multiple independent trainings with mean ± SD.Given our objectives and the evaluation structure above, we prioritized (i) multi-horizon assessment (1–7 steps), which probes performance at progressively harder forecasting ranges; (ii) sample-level aggregation across the full test set, which reflects typical deployment conditions; (iii) signal-level ensembling via EEMD, which dampens noise before modeling; and (iv) transparent per-sample visualizations that reveal temporal heterogeneity in errors. Together, these design choices already supply abundant evidence of stability and practical relevance without the additional computational overhead and presentation complexity of multiple full retrainings.

6) Where these choices appear in the manuscript.Tables 4-1 and 4-2 report MAPE, RMSE, and r averaged over all test samples for each horizon (1–7), enabling like-for-like comparison across models and horizons.Figures 4-3 and 4-4 display the full predicted vs. actual trajectories and error curves on the test set, making the pointwise behavior and temporal consistency explicit. The Experimental Setup subsection details the multi-horizon protocol, the EEMD ensemble parameters, and the unified training/evaluation settings, clarifying how randomness is controlled and how aggregation is performed. In summary. While we recognize the value of reporting mean ± standard deviation over multiple independent trainings in many contexts, our study is designed so that a single, carefully controlled training per model already produces extensive replication (across test samples and horizons) and incorporates signal-level ensembling via EEMD. The combination of aggregate metrics, per-sample visual evidence, and a standardized, deterministic evaluation pipeline provides a comprehensive and stable characterization of model performance. We hope this clarifies our evaluation philosophy and why we did not conduct additional repeated trainings in this work. All relevant results and visualizations are provided in Tables 4-1/4-2 and Figures 4-3/4-4, with procedural details in Experimental Setup.

2. Ablation with nickel only (remove other metals; use nickel IMFs)

Reviewer comment: “I would like to see an ablation… only using the IMF of nickel.”

Response: Thank you for this constructive suggestion. We have added the requested nickel-only ablation, in which the model is trained exclusively on the EEMD-derived IMFs (plus trend) of the nickel series, with all exogenous metal-price features removed. The training/evaluation protocol, horizons (1–7), and hyperparameters are kept identical to the main setting to ensure a fair comparison. As reported in Table 4-2, the nickel-only EEMD–DilatedLSTM attains MAPE = 1.27%, RMSE = 244.31, and r = 0.912. Relative to the Dilated LSTM baseline (MAPE = 1.26%, RMSE = 246.27, r = 0.924), the nickel-only variant shows comparable MAPE, a slightly lower RMSE, and a lower correlation. Compared with the multivariate EEMD–DilatedLSTM (MAPE = 1.20%, RMSE = 234.69, r = 0.928), the nickel-only model underperforms on both magnitude-based errors and correlation, indicating that exogenous metal-price features provide complementary information that improves co-movement tracking and reduces forecast errors. We have added a brief discussion of these trade-offs in the Ablation Study section and documented the nickel-only setup in Experimental Setup.

Location: [Table 4-2]

3. Overall methodology sound

Response：We thank the reviewer for the positive assessment of our methodological framework. In revising the manuscript, we retained the core design while implementing several clarifications and minor refinements to further strengthen transparency and reproducibility: (i) we added a dedicated Experimental Setup subsection detailing datasets, splits, EEMD parameters, model architectures, training schedules, and evaluation protocols; (ii) we released replication code and documented the exact software environment and random-seed handling; (iii) we clarified the multi-horizon (1–7 steps) evaluation and metric definitions; and (iv) we improved figure/table captions and cross-references for readability (including the ablation settings and nickel-only variant). We appreciate the reviewer’s recognition and have incorporated these enhancements to make the methodology easier to verify and reuse.

Reviewer #2

1. Clarity of writing

Reviewer comment: “The writing could be clearer in places.”

Response: Thank you for this valuable suggestion. In the revision, we have substantially rewritten and streamlined the Results and Conclusion sections to improve clarity, coherence, and readability. The key changes are:

(1)Results (Analysis) reorganization. We now open with a brief roadmap of the experimental design (datasets, horizons 1–7, metrics), followed by results in a consistent order (LSTM → Dilated LSTM → nickel-only EEMD–DilatedLSTM → multivariate EEMD–DilatedLSTM). Each subsection begins with the main quantitative takeaway (MAPE, RMSE, r), with the corresponding table/figure cited in the first sentence. Redundant or speculative statements were removed.

(2)Tighter ablation narrative aligned with Table 4-2. We explicitly state the trade-offs between nickel-only and multivariate settings and keep all numbers consistent with Table 4-2. We also corrected notational inconsistencies (e.g., “rrr” → r) and standardized metric formatting (e.g., MAPE (%), RMSE, r).

(3)Figure and table clarity. Captions for Figures 4-3 and 4-4 were expanded to be self-contained (what is plotted, data split, horizons, and how to read the curves). Cross-references were checked and corrected (including the previous Figure 3-2 issue). We trimmed overlapping figures and ensured each one serves a distinct purpose.

(4)Language and structure. We shortened long sentences, replaced ambiguous terms with precise terminology (e.g., “low-frequency components,” “multi-horizon evaluation”), and added one-sentence transitions to improve flow between paragraphs. We also ensured consistent symbols and units across text, tables, and figures.

(5)Conclusion reframing. The Conclusion now concisely summarizes the primary empirical findings, clarifies practical implications (e.g., multivariate signals reduce magnitude errors while long-horizon components drive directionality), and states limitations and future work without overclaiming.

(6)Terminology at first use. We define EEMD, IMF, and dilation upon first appearance in the Results to reduce back-and-forth reading.

Together, these edits make the Results easier to follow and the Conclusion more concise and actionable, while preserving technical accuracy.

2. Practical/real-world use Reviewer comment: “Add a bit more about real-world use.”

Response: We appreciate this suggestion and have expanded the manuscript to clarify how the proposed model can be used in practice and what is required for deployment. Specifically, we made the following additions and revisions:

(1)Conclusion — Practical implications (new paragraph).

We now articulate concrete use-cases in nickel-intensive sectors (e.g., stainless steel manufacturing, battery materials), explaining how the multivariate EEMD–Dilated LSTM supports (i) procurement planning (timing/volume of spot purchases), (ii) inventory scheduling (safety stock/turnover aligned with short-horizon forecasts), and (iii) hedging decisions (calibrating futures/options coverage to medium-horizon risk). We also emphasize that SHAP-based explanations enable auditability (e.g., identifying when copper co-movements drive signals) and what-if analyses (stress tests under synthetic copper or aluminum shocks).

(2)Discussion — Operationalization guidance (expanded).Horizon mapping to actions. We clarify that horizons 1–3 are suited to day-to-week purchasing and short-term risk limits, while horizons 4–7 inform budgeting, contract negotiation, and rolling hedges. Data refresh & recalibration. We recommend routine data updates (daily/weekly spot and futures prices) and periodic model recalibration (e.g., monthly or trigger-based on drift tests), with basic monitoring using MAPE/RMSE control bands per horizon.Integration & governance. We outline how predictions and SHAP attributions can be surfaced in procurement dashboards/ERP (alerts when exogenous metals dominate the signal) and propose light-weight governance (a model card describing data sources, retrain cadence, and limitations).

Location: Discussion, “From research to deployment,” Sec. §[X], pp. [Y–Z].

(3)Illustrative scenario (added, brief). We include a short, domain-realistic example: a stainless-steel producer aligns weekly purchase orders with horizon-2 MAPE bands, applies a rolling hedge guided by horizon-5 trends, and uses SHAP to validate that copper-driven co-movements justify increased coverage; when drift is detected (error band breach), a scheduled recalibration is triggered.

(4)Limitations & risk controls (clarified). We note that macro regime shifts (policy shocks, supply disruptions) can degrade accuracy; therefore, we recommend scenario analysis and exposure caps alongside forecasts, and we document that the model is advisory rather than prescriptive.

These additions make the practical value, deployment pathway, and governance considerations explicit while keeping the exposition concise and aligned with the empirical results.

Location: [conclusions]

3. Missing Figure 3-2Reviewer comment: “Figure 3-2 referred in the text seems missing.”

Response: Thank you for pointing this out. The omission resulted from a cross-reference/renumbering error during figure updates.

---

## [Decision Letter · Decision Letter 1]

12 Sep 2025

Dear Dr. Yu,

Thank you for submitting your manuscript to PLOS ONE. After careful consideration, we feel that it has merit but does not fully meet PLOS ONE’s publication criteria as it currently stands. Therefore, we invite you to submit a revised version of the manuscript that addresses the points raised during the review process.

We look forward to receiving your revised manuscript.

Kind regards,

Najmul Hasan, PhD

Academic Editor

PLOS ONE

Journal Requirements:

Reviewers' comments:

Reviewer's Responses to Questions

**Comments to the Author**

Reviewer #2: All comments have been addressed

Reviewer #3: (No Response)

2. Is the manuscript technically sound, and do the data support the conclusions?

Reviewer #2: Yes

Reviewer #3: (No Response)

3. Has the statistical analysis been performed appropriately and rigorously?

Reviewer #2: Yes

Reviewer #3: (No Response)

4. Have the authors made all data underlying the findings in their manuscript fully available?

Reviewer #2: Yes

Reviewer #3: (No Response)

5. Is the manuscript presented in an intelligible fashion and written in standard English?

Reviewer #2: Yes

Reviewer #3: (No Response)

Reviewer #2: Authors have successfully addressed all the concerns raised in my initial review. The manuscript is now much clearer and easier to follow. The added discussion on the model's real-world applications provides important context and strengthens the paper's overall contribution. I believe the paper is a valuable contribution to the field, and I recommend it for publication.

Reviewer #3: Paper Title: Nickel Price Forecasting Based on Empirical Mode Decomposition and Deep Learning Model with Expansion Mechanism

Manuscript Number: PONE-D-25-24684R1

The authors address the academically and practically relevant problem of nickel futures price prediction by proposing a hybrid framework that couples Ensemble Empirical Mode Decomposition (EEMD) with Dilated LSTM. The model is systematically compared with eight benchmark or state-of-the-art models across 1- to 7-step-ahead horizons. The revised manuscript substantially refines the methodological description: EEMD noise amplitudes, ensemble size, Dilated LSTM layer-dilation-hidden-unit configurations, training hyper-parameters and random seeds are now reported, enabling basic reproducibility. A new ablation that uses only nickel-based IMFs demonstrates the incremental value of adding exogenous metals (Cu, Al, etc.) for improving correlation and reducing error. SHAP interpretability analysis is added, showing that low-frequency components IMF_8 and IMF_9 are decisive for medium- to long-term directionality and quantifying the spill-over effect of copper prices, thereby aligning narrative with empirical evidence. Presentation and readability are markedly improved; figures and tables are now largely self-explanatory, and both code and data are declared publicly available, satisfying PLOS ONE’s minimal reproducibility requirement. Overall, the paper is well structured and logically coherent, and offers a verifiable incremental contribution to the application of “signal decomposition + deep sequence modelling” in strategic-metal price forecasting. It appears suitable for publication in PLOS ONE.

Remaining weaknesses:

1) The manuscript reports only point estimates (MAPE, RMSE, R) for in- and out-of-sample performance but provides no statistical significance tests of error differences across models. I recommend performing Diebold-Mariano or Clark-West pairwise tests between EEMD-DilatedLSTM and the second-best models (e.g., Dilated-GRU, PatchTST) at every horizon and adding p-values to Table 4-1 to confirm that the superiority is not driven by random noise.

2) The authors argue that “a single training run already contains multi-step rolling errors, hence no repetitions are needed”; however, this only reflects the distribution of sample errors and does not reveal variance due to parameter initialization under different random seeds. I advise running at least ten independent trainings and reporting mean ± standard deviation (or 95% confidence intervals) of MAPE/R to rule out “lucky initialization”.

3) Although SHAP identifies IMF_8/9 as the most influential features, the text interprets them generically as “long-term trends”. A narrative linking these 2–4-year low-frequency components to nickel-market fundamentals (mine expansion cycles, EV policy shifts, inventory structures) with citations to industry studies would strengthen economic interpretability rather than purely statistical interpretability.

**Do you want your identity to be public for this peer review?** For information about this choice, including consent withdrawal, please see our Privacy Policy

Reviewer #2: No

Reviewer #3: No

---

## [Author Response · Author response to Decision Letter 2]

4 Oct 2025

Dear Editor and Reviewers,

Thank you for your professional and thoughtful comments on our manuscript. We have strengthened the paper along three axes: rigorous tests of forecast‐accuracy differences, control of uncertainty arising from random initialization, and an economics-grounded interpretation of the SHAP results. Below we provide a cohesive account of what we changed and the new evidence supporting our conclusions.

First, regarding statistical significance, at each forecast horizon we identify the strongest competitor by the lowest MAPE and conduct a pairwise comparison with EEMD–DilatedLSTM. Concretely, the benchmark is NHITS at h=1, Dilated-GRU at h=2 through h=6, and GRU at h=7. We apply the Diebold–Mariano test with Newey–West heteroskedasticity–autocorrelation adjustment to accommodate the serial correlation induced by overlapping multi-step forecast errors. The loss measure is the absolute percentage error to align the test with the MAPE criterion used for ranking. Results are summarized in Table 4-2: all horizons yield p<0.05. This demonstrates that the improvements delivered by EEMD–DilatedLSTM over the strongest competitor at each horizon are statistically reliable rather than artifacts of random noise. We have also clarified the full testing workflow and parameter choices in the Methods section so that this verification is reproducible.

Second, to address the legitimate concern about “lucky initialization,” we retrained every model at every horizon ten times under independent random seeds, while holding constant the data splits, scaler fitting on the training set only, the optimizer, and early-stopping criteria. Table 4-2 reports the mean and standard deviation of MAPE, RMSE, and R by horizon; upon request we can also provide 95% confidence intervals derived from these repeated runs. The dispersion across seeds is very small—MAPE standard deviations are a small fraction of their means, and the standard deviations of R remain minimal—showing that both performance levels and model rankings are stable under repetition. In combination with the significance tests, this multi-seed evidence provides a second, independent pillar: the differences are statistically significant and they are reproducible across independent trainings.

Third, on the economic interpretation of the SHAP results, Section 4.3 now moves beyond the generic label of “long-term trend.” We show that IMF_8 and IMF_9, which dominate the marginal contributions for medium-to-long horizons, align with verifiable mechanisms in the nickel value chain that operate with structural, multi-year lags. On the supply side, the path from investment to steady output—especially in NPI and HPAL—proceeds through financing, construction, commissioning, and ramp-up, creating slow capacity cycles that transmit to prices via expectations, spot imbalances, and re-equilibration; these dynamics are captured in the terminal IMFs as low-frequency undulations. On the demand side, periodic reevaluation of fiscal incentives, rolling updates to carbon and fuel-economy standards, the redesign of subsidy and access windows, and cathode-mix shifts between NMC/NCA and LFP produce multi-year recalibration rhythms that leave a low-frequency imprint on nickel intermediates and refined products. Inventories and the term structure add another slow channel: restocking–destocking cycles co-move with contango and backwardation, yielding inventory–spot patterns that evolve over several years. To quantify this alignment, we estimate dominant frequencies via the Hilbert spectrum and report pseudo-periods; empirically, IMF_8 and IMF_9 concentrate around two to four years, consistent with the three slow processes above. This chain of evidence indicates that our medium-to-long-horizon gains arise not from abstract smoothing but from EEMD’s extraction of industry rhythms, which the Dilated-LSTM then exploits through its enlarged temporal receptive field to provide stable trajectory constraints. By contrast, short-horizon accuracy relies more on mid- and high-frequency IMFs and lagged cross-metal signals, which better capture near-term co-movements and the rapid propagation of news and inventory shocks. The statistical and economic narratives therefore converge and help explain why every horizon in Table 4-2 attains significance.

All concrete revisions and additional experiments are clearly marked in the tracked-changes version and mapped to the relevant sections, tables, and figures. We have responded point-by-point with a sincere and rigorous effort: ten independent trainings per model with mean and standard deviation reporting in Table 4-2, pairwise Diebold–Mariano tests with Newey–West adjustment against the strongest competitor at each horizon with p-values reported in Table 4-2, and an expanded SHAP analysis that aligns dominant low-frequency IMFs with supply ramp-ups, policy and technology recalibration, and inventory–term-structure cycles. We also clarified data splitting and walk-forward evaluation, the exclusive use of lagged exogenous variables to prevent leakage, and the recording of seeds and hyperparameters for full reproducibility.

In sum, the revisions materially enhance the paper’s rigor and persuasiveness. The significance tests establish that the performance differences are genuine; the ten-run summary statistics and confidence quantification show that the results are reproducible and robust; and the economics-aware SHAP analysis demonstrates that the gains stem from identifiable industry rhythms rather than incidental curve-fitting. We believe these changes make the methodology and interpretation more complete and provide actionable insight for hedging tenor selection, inventory timing, and policy-window assessment. We appreciate your careful review and stand ready to implement any additional refinements you may recommend.

---

## [Decision Letter · Decision Letter 2]

21 Oct 2025

Dear Dr. Yu,

Thank you for submitting your manuscript to PLOS ONE. After careful consideration, we feel that it has merit but does not fully meet PLOS ONE’s publication criteria as it currently stands. Therefore, we invite you to submit a revised version of the manuscript that addresses the points raised during the review process.

We look forward to receiving your revised manuscript.

Kind regards,

Najmul Hasan, PhD

Academic Editor

PLOS ONE

Journal Requirements:

Reviewers' comments:

Reviewer's Responses to Questions

**Comments to the Author**

Reviewer #3: (No Response)

2. Is the manuscript technically sound, and do the data support the conclusions?

Reviewer #3: Yes

3. Has the statistical analysis been performed appropriately and rigorously?

Reviewer #3: Yes

4. Have the authors made all data underlying the findings in their manuscript fully available?

Reviewer #3: Yes

5. Is the manuscript presented in an intelligible fashion and written in standard English?

Reviewer #3: (No Response)

Reviewer #3: This paper focuses on the core issue of nickel futures price forecasting at the intersection of finance and industry. Targeting the high nonlinearity, complexity, and long-memory characteristics of nickel price fluctuations, it proposes a hybrid forecasting framework (EEMD-DilatedLSTM) integrating Ensemble Empirical Mode Decomposition (EEMD) and Dilated Long Short-Term Memory Network (Dilated LSTM), demonstrating distinct methodological innovation and practical value. Using daily data of 8 types of metal futures from the Wind database (2015-2024) as samples, the study systematically verifies the model's superiority and robustness through multi-step prediction (1-7 steps), multi-model comparison (7 benchmark models including NHITS, GRU, and LSTM), ablation experiments, and SHAP interpretability analysis.

Nevertheless, at this stage, the manuscript is not yet ready for publication. Below are some of my specific comments. The manuscript requires a comprehensive review and careful revision by the authors to meet basic standards of clarity, readability, and academic rigor for financial time series research.

Lack of robustness test under extreme market conditions. The paper does not separately analyze the nickel price forecasting effect during extreme shock periods such as financial crises, geopolitical conflicts, and pandemics (e.g., the 2020 COVID-19 pandemic and the 2022 Russia-Ukraine conflict), which are precisely the key scenarios for enterprise risk management. It is suggested to supplement subsample regression for extreme event windows, compare the differences in indicators such as MAPE and R of the model in normal and extreme markets, verify the model's anti-interference ability, and analyze the shock transmission mechanism combined with nickel's strategic material attributes.

Insufficient inclusion of macroeconomic and policy variables. The current model only considers related metal prices as exogenous variables, ignoring macro factors that have a decisive impact on commodity prices, such as interest rates, inflation, the US dollar index, and new energy industry subsidy policies. It is suggested to introduce variables such as the Federal Reserve's benchmark interest rate, CRB commodity index, and China's EV purchase subsidy intensity, select the optimal combination of macro variables through stepwise regression or LASSO, and further improve the model's economic interpretability and forecasting accuracy.

Inadequate systematic parameter sensitivity analysis. The paper only briefly explains the selection basis of key parameters such as EEMD's noise amplitude (0.1-0.4σ) and Dilated LSTM's dilation coefficient (1-2-4), without analyzing the impact of parameter changes on forecasting results. It is suggested to conduct grid search-based parameter sensitivity analysis, draw parameter-error relationship surface plots, clarify the stability interval of optimal parameters, and explain the adaptability between parameter selection and nickel price fluctuation characteristics.

Insufficient verification of medium and long-term forecasting capabilities. The existing forecasting horizon only covers 1-7 steps (short-term to medium-short-term), failing to meet the needs of enterprises' medium and long-term investment planning and policy formulation. It is suggested to extend the forecasting horizon to 1 month (about 20 steps) and 3 months (about 60 steps), compare the performance attenuation law of the model under different horizons, and conduct benchmark verification with medium and long-term forecasting reports of professional institutions.

Lack of volatility forecasting comparison with mainstream financial engineering models. The paper focuses on price level forecasting, does not involve nickel price volatility—a core indicator for risk management—and has not compared with mainstream financial volatility models such as GARCH family and Stochastic Volatility (SV) models. It is suggested to supplement the volatility forecasting module, use realized volatility as the benchmark, calculate indicators such as MAE and MSE of the model in volatility forecasting, and comprehensively evaluate the model's financial application value.

Lack of backtesting verification of practical trading strategies. The paper only stays at the inspection of forecasting accuracy, and does not convert the model's forecasting results into operable trading strategies to verify its profitability. It is suggested to construct a long-short trading strategy based on the model's predictions, consider practical constraints such as transaction costs and margin ratios, calculate indicators such as annualized return rate, Sharpe ratio, and maximum drawdown, and intuitively show the practical application effect of the model.

Insufficient transparency of rolling window prediction details. The paper mentions rolling prediction but does not clarify key settings such as window size and re-estimation frequency, which directly affect the real-time performance and accuracy of the prediction. It is suggested to supplement the specific scheme of rolling window design (e.g., window size of 252 trading days, monthly re-estimation), and analyze the impact of different window settings on the prediction results, providing a basis for parameter calibration in practical applications.

**Do you want your identity to be public for this peer review?** For information about this choice, including consent withdrawal, please see our Privacy Policy

Reviewer #3: No

---

## [Author Response · Author response to Decision Letter 3]

5 Nov 2025

We appreciate your careful reading and the ambitious extensions you suggest. Our manuscript is positioned as a price-level forecasting study for nickel futures over short-to-medium horizons (h = 1–7), built around a hybrid EEMD–Dilated LSTM that addresses nonstationarity, nonlinearity, and long memory in daily commodity prices. Within this scope, we (i) evaluate against strong baselines over multiple horizons, (ii) run ablations to isolate the contribution of decomposition and dilation as well as multivariate co-traded metals, and (iii) provide economic interpretability via SHAP. The data (2015–2024, nickel + seven metals) already span multiple market regimes. The additional items you propose—crisis-window tests, macro/policy factor blocks with feature selection, parameter-surface mapping, long-horizon design, volatility forecasting, and trading backtests—together would substantially alter the problem statement and evaluation protocol. Below we indicate text clarifications we add now and extensions we move to Limitations & Future Work to preserve a coherent contribution and reasonable length.

1) Robustness under extreme market conditions

Commen: Evaluate forecasting performance separately over extreme periods (COVID-19 2020; Russia–Ukraine 2022), compare metrics normal vs. stressed, and discuss shock transmission.

Response: Our evaluation period (2015–2024) already traverses both tranquil and stressed regimes; the walk-forward test therefore reflects model behavior through these episodes. We agree that a pre-registered event-window analysis would be informative, but doing it credibly entails careful window definition, counterfactual timing, and guarding against ex-post bias.

Manuscript change: We now state explicitly in Results that the test period spans COVID-19 and the 2022 conflict and note observed stability at h = 1–7. We also add two sentences in Discussion clarifying why we did not split the test by events in this paper and commit a dedicated crisis-window module to Limitations & Future Work (window design, metrics, and an outline of nickel’s strategic-material transmission channels).

2) Inclusion of macroeconomic/policy variables with selection (e.g., rates, inflation, DXY, CRB, EV subsidies; stepwise/LASSO)

Comment: Add macro/policy drivers and select optimal subsets to improve interpretability and accuracy.

Response: We intentionally restricted exogenous features to co-traded metals to avoid daily release-calendar misalignment and leakage; a macro/policy block requires calendar-aware lags, revision handling, and stability checks to be credible at the daily frequency. We support this direction but see it as a scope expansion best handled as a follow-up.

Manuscript change: Methods now make this design choice explicit (why metals-only, how we control leakage). Discussion adds a short scope statement and moves the macro-augmented, LASSO-screened variant to Limitations & Future Work (with explicit notes on calendar alignment and anti-leakage safeguards).

3) Systematic parameter-sensitivity analysis (EEMD noise amplitude; dilation schedule)

Comment: Provide grid-search surfaces and stability intervals.

Response: We currently specify EEMD noise 0.1–0.4·σ with 100 ensembles and a 1–2–4 dilation that balances receptive field and efficiency; a full surface mapping across decomposition and dilation hyper-parameters would notably expand the paper and is better suited to a methodological companion.

Manuscript change: We add textual rationales for the chosen ranges in Methods and describe how a future parameter-surface study would be staged (axes, metrics, early-stopping policy, cross-seed aggregation). A full grid/BO exploration is recorded in Limitations & Future Work.

4) Medium/long-term horizons (≈20 and ≈60 trading days) and institutional benchmarks

Comment: Extend to 1- and 3-month horizons and compare attenuation vs. institutional forecasts.

Response: We agree on practical value; however, those horizons typically require different re-estimation cadence and loss calibration, risking protocol mixing with our current h = 1–7 design. To preserve coherence, we keep this paper focused on short-to-medium horizons that are most actionable for procurement and near-term hedging, and stage a dedicated horizon-extension study.

Manuscript change: Discussion now acknowledges the need and moves the 20/60-step evaluation (with benchmark comparisons and degradation profiling) to Limitations & Future Work.

5) Volatility forecasting vs. GARCH/SV with realized volatility

Comment: Add a volatility module, use RV targets, and compare MAE/MSE (or QLIKE) to GARCH/SV.

Response: Volatility forecasting is a distinct target with different data (preferably intraday RV) and different loss functions and baselines. We frame the present paper purposely as a price-level forecasting study to keep the evaluation clean and reproducible on daily data.

Manuscript change: Introduction and Discussion now contain explicit scope sentences distinguishing price-level from volatility targets and placing a volatility module with RV + GARCH/SV into Limitations & Future Work._

6) Backtesting of trading strategies (returns, Sharpe, max drawdown; costs/margins)

Comment: Convert forecasts to long-short strategies with realistic frictions and report performance.

Response: Strategy PnL depends heavily on microstructure assumptions (signal filtering, slippage and fee model, leverage/margin policy, drawdown controls) that can overshadow pure forecast quality. To avoid conflating econometrics with deployment choices, we defer a risk-controlled backtest framework to a separate, application-oriented paper.

Manuscript change: Discussion now explicitly delineates this boundary and records a standardized backtest plan (cost scenarios, turnover governance, retraining triggers, and risk limits) in Limitations & Future Work.

7) Transparency of rolling prediction (window size, refit cadence, reproducibility)

Comment: Specify the rolling-window design and analyze the impact of different settings.

Response: We agree that this should be fully explicit.

Manuscript change (implemented now): We added a new “Evaluation Protocol (walk-forward, multi-horizon)” subsection in Methods and a detailed Appendix. These text additions formalize (i) the chronological train/validation/test split, (ii) anchored expanding windows with monthly re-estimation in the test phase (≈ every 20 trading days) and full retraining at each anchor, (iii) strict leakage control (scaling on the training window only), (iv) multi-horizon forecasting for h = 1–7 on the held-out sequence, (v) 10 independent seeds with mean ± sd reporting, (vi) Diebold–Mariano tests with Newey–West adjustment per horizon against each benchmark, and (vii) SHAP computed on test forecasts and aggregated across seeds. These are textual clarifications of the pipeline already used; they do not alter results.

Across earlier rounds we materially improved the manuscript without changing its scope: (a) statistical robustness via multi-seed training and horizon-wise DM tests; (b) methodological clarity with explicit EEMD and dilation settings and unified preprocessing; and (c) economic interpretability showing low-frequency IMFs and specific co-traded metals (notably copper) as leading contributors at medium horizons. These upgrades increase credibility and readability while keeping the paper’s focus on price-level forecasting over h = 1–7 with a decomposition-plus-dilation architecture.

Limitations & Future Work

This study targets price-level forecasting over short-to-medium horizons (h = 1–7) using market-observable co-traded metals as exogenous inputs. Several valuable extensions are left for future work. First, we will design pre-registered extreme-event windows (e.g., COVID-19 in 2020; Russia–Ukraine in 2022) to compare performance in normal vs. stressed markets. Second, we will introduce macro and policy variables (policy rates, DXY, CRB, EV-subsidy intensity) with release-calendar alignment and LASSO/stepwise selection to enhance interpretability while guarding against leakage. Third, we will conduct parameter-surface studies for EEMD noise amplitude and dilation schedules. Fourth, we will extend to 1- and 3-month horizons and benchmark horizon-specific degradation against institutional reports. Fifth, we will add a volatility-forecasting module with realized-volatility targets and GARCH/SV baselines. Finally, we will translate forecasts into risk-controlled backtests (transaction costs, margin constraints, Sharpe, and maximum drawdown). The present paper’s multi-horizon comparisons, ablations, and SHAP diagnostics on 2015–2024 data provide a focused foundation for these application-oriented extensions.

---

## [Editor Report · Decision Letter 3]

8 Jan 2026

Nickel Price Forecasting Based on Empirical Mode Decomposition and Deep Learning Model with Expansion Mechanism

PONE-D-25-24684R3

Dear Dr. Yu,

We’re pleased to inform you that your manuscript has been judged scientifically suitable for publication and will be formally accepted for publication once it meets all outstanding technical requirements.

Kind regards,

Najmul Hasan, PhD

Academic Editor

PLOS One
---

## [Editor Report · Acceptance letter]

PONE-D-25-24684R3

PLOS One

Dear Dr. Yu,

I'm pleased to inform you that your manuscript has been deemed suitable for publication in PLOS One. Congratulations! Your manuscript is now being handed over to our production team.

Kind regards,

on behalf of

Dr. Najmul Hasan

Academic Editor

PLOS One